# Interrogations of single-cell RNA splicing landscapes with SCASL define new cell identities with physiological relevance

Xianke Xiang[1,2], Yao He[3], Zemin Zhang [3,4] & Xuerui Yang [1,2] ✉

RNA splicing shapes the gene regulatory programs that underlie various physiological and disease processes. Here, we present the SCASL (single-cell clustering based on alternative splicing landscapes) method for interrogating the heterogeneity of RNA splicing with single-cell RNA-seq data. SCASL resolves the issue of biased and sparse data coverage on single-cell RNA splicing and provides a new scheme for classifications of cell identities. With previously published datasets as examples, SCASL identifies new cell clusters indicating potentially precancerous and early-tumor stages in triple-negative breast cancer, illustrates cell lineages of embryonic liver development, and provides fine clusters of highly heterogeneous tumor-associated CD4 and CD8 T cells with functional and physiological relevance. Most of these findings are not readily available via conventional cell clustering based on single-cell gene expression data. Our study shows the potential of SCASL in revealing the intrinsic RNA splicing heterogeneity and generating biological insights into the dynamic and functional cell landscapes in complex tissues.

Variations in RNA splicing serve as a major cause of extensive transcriptome complexity in mammalian species[1–3]. Alternative splicing (AS) is closely related to various cellular activities such as cell lineage differentiation, cell proliferation, and apoptosis[4–8]. Shifted landscapes of RNA splicing are primary outcomes as well as key driving factors of various physiological and pathological processes[9,10]. For example, complexity in RNA splicing is a main characteristic of tumor heterogeneity related to metastasis, drug sensitivity, and relapse, thereby serving as a prognostic marker for multiple cancer types, including breast cancer[11], glioblastoma[12], non-small cell lung cancer[13], and ovarian cancer[14].

Transcriptome profiling at single-cell resolution has become a common practice for dissection of cell heterogeneity in complex tissues[15,16]. Single-cell transcriptome profiling based on the SMART-seq method generates RNA-seq reads with coverage of full-length transcripts, rather than just the 3' regions, as do most droplet-based methods such as Drop-seq and 10x Genomics. Therefore, the

RNA splicing profiles are readily available from these datasets. However, the vast majority of previous studies employing single-cell RNA-seq (scRNA-seq) have solely relied on the overall expression levels of genes for classifications of cell subpopulations[17]. The RNA splicing landscapes, which represent a critical layer of transcriptome regulation, have not been considered in most of the previous studies.

Unlike single-cell gene expression profiles, AS profiles represent the relative abundance of different transcript isoforms, thereby sheltering the intrinsic fluctuations of single-cell gene expression levels. Importantly, even though bulk RNA-seq data of heterogeneous tissues or cells commonly shows mixtures of isoforms, individual cells are frequently dominated by specific isoforms for many genes[9], indicating strong cell specificity of AS events. Finally, in many cases, AS results in different protein products, rather than just changes in protein abundance, therefore potentially leading to more dramatic shifts in biological pathways and processes. These features make single-cell AS

[1]MOE Key Laboratory of Bioinformatics, School of Life Sciences, Tsinghua University, Beijing 100084, China. [2]Center for Synthetic & Systems Biology, Tsinghua University, Beijing 100084, China. [3]Biomedical Pioneering Innovation Center and School of Life Sciences, Peking-Tsinghua Center for Life Sciences, Academy for Advanced Interdisciplinary Studies, Peking University, Beijing 100871, China. [4]Cancer Research Institute, Shenzhen Bay Lab, Shenzhen 518132, China. ✉e-mail: yangxuerui@tsinghua.edu.cn

profiles promising indexes for the classification of single cells with potentially high physiological relevance.

The existing approaches for quantifying alternative splicing (AS) levels were based on either PSI (percent splice-in) or junction reads. Most of these AS analysis tools primarily focus on identifying differential splicing events. For instance, BRIE[18] (for single-cell data) and Expedition[19] (for single-cell data) employ Bayesian models to estimate PSI for differential splicing analysis, while rMATS[20] (for bulk data) employs a linear mixed-effects model. Psix[21] quantifies AS using PSI values and identifies splicing events associated with cell state through autocorrelation models. On the other hand, Leafcutter[22] identifies junction reads for definitions of AS events and then adopts a generalized linear model to quantify the AS events. FRASER[23] uses a different strategy for identifying and quantifying the AS events from junction reads. MARVEL[24] relies on PSI defined by rMATS and junctions reads to define single-cell AS and then integrates gene expression data to analyze the effects of differential splicing between cell groups. Additionally, newer methods have emerged that define AS levels at the gene level rather than for specific AS events, such as SpliZ[25]. Nonetheless, these tools typically require known cell labels for differential analysis, and methods specifically designed for unsupervised clustering based on single-cell splicing landscapes are urgently needed.

In the present study, we developed SCASL (single-cell clustering based on alternative splicing landscapes), a new strategy of cell clustering based on single-cell RNA splicing landscapes. Specifically, SCASL employs a strategy similar to that of LeafCutter[22] and FRASER[23] to identify AS events from the junction reads in single-cell SMART-seq data. This method does not rely on predefined annotation of transcriptomes, thereby recovering both known and novel AS events. AS probabilities are then inferred from proportions of the junction reads assigned to the same 5′ AS or 3′ AS events. The profiles of AS probabilities are then used for imputations of the missing values, followed by spectral clustering of the cells.

We tested the performance of SCASL on defining cell subpopulations with single-cell RNA-seq data of embryonic tissue development, tumor cells, and tumor-associated immune cells. Cell lineage differentiation is extremely important in physiological and disease development. However, owing to the intrinsic gene expression fluctuations in the progressive process of lineage differentiation, it has been challenging to clearly define cells at key transitional stages simply based on gene expression profiles. In addition, highly heterogeneous cells in complex tissues, such as tumor cells and tumor-associated immune cells, usually show very diverse gene expression profiles, making it difficult to clearly and consistently define boundaries between cell clusters.

When applied to datasets of triple negative breast cancer, SCASL defined clear cell subtypes indicating precancerous transformation of epithelial cells and early-stage tumor cells. With embryonic liver data, SCASL recovered a series of transitional stages during developments of the hepatocyte and cholangiocyte lineage lineages. Finally, highly diverse tumor associated T cells were also classified into subpopulations with distinct molecular and cellular characteristics related to tumor infiltration, activation, and exhaustion. In contrast, the methods commonly used with single-cell gene expression data did not recapitulate such cell groups with physiological relevance. Therefore, the heterogeneous landscapes of RNA splicing provide unique information for precise definitions of physiologically relevant cell subtypes, which is valuable for understanding the fine tuning that occurs during complex development and disease contexts. We propose that SCASL can be used as an efficient method for mining such information.

## Results

### The SCASL pipeline for cell clustering based on single-cell RNA splicing landscapes

The SCASL pipeline consists of three major steps, i.e., (i) establishment of a single-cell RNA AS probability matrix, (ii) imputation of the missing values in the matrix, and (iii) spectral clustering of the single cells based on the patched AS probability matrix (details in the Methods section). In brief, first, de novo identification of RNA splicing events is performed by mapping single-cell RNA-seq reads, which are usually obtained from SMART-seq techniques, to the reference genome. This procedure does not rely on prior annotation of exons, thereby capturing complex AS events in a more comprehensive manner. Multiple splicing schemes taking place with a common 5′ or 3′ splice site are then clustered together, and the split reads are counted to estimate the probabilities of these different 3′ AS or 5′ AS events (Fig. 1). This procedure is performed for each single cell, generating an AS probability matrix representing the transcriptome-wide landscape of splicing at single-cell resolution.

Due to the relatively low sequencing depth and limited coverage of single-cell RNA-seq data, the probabilities of many AS events cannot be reliably estimated for all the single cells in a dataset. Therefore, the AS probability matrix greatly suffers from frequent missing values (marked as NA). SCASL introduces a strategy of iterative weighted KNN for imputation of these missing values (Fig. 1). In brief, SCASL determines $k$ nearest neighbors for each cell based on the Euclidean distances between the AS probabilities of the cells. The missing values of each cell are then inferred by taking the weighted averages of the neighboring cells. The new AS probability matrix is then used to update the missing values inferred from the previous round. This process is conducted iteratively until convergence. We chose KNN because it significantly outperformed other imputation methods, such as mean value imputation and two model-dependent methods, MAGIC[26] and scImpute[27] (data not shown).

The imputation strategy was tested with forced dropouts of different proportions of the non-NA values in 3 datasets. The imputed AS probabilities were highly accurate compared to the true values (Fig. S1A). Even if 90% of the AS probability values were dropped-out, the majority of these data points were still correctly recovered. This suggests that transcriptome-wide splicing information is highly redundant for identifying cells with similar splicing landscapes, thereby allowing precise imputation of missing AS probabilities.

Despite the high fidelity of the imputation process, potential errors of the AS probabilities inferred for the originally missing values still cannot be completely ruled out. In addition, the sparsity pattern of the AS matrix, which is remotely related to gene expression, should also be informative for the cellular physiological status, as suggested by previous studies[3,28]. Therefore, new binary vectors marking the positions of the imputed values are concatenated with the vectors of AS probabilities, giving rise to a new and expanded AS probability matrix.

SCASL employs spectral clustering, a graph-based clustering algorithm[29], with the modified AS probability matrix above. The elbow method is implemented to help determine the optimal number of clusters[30]. To showcase its performance, SCASL was applied to 6 sets of single-cell transcriptome data generated by the SMART-seq method (Table 1). The resulting cell clusters were significantly different than those obtained simply from the RNA expression profiles with the canonical Seurat[31] pipeline (Fig. S1B), which will be discussed in the following sections.

We then tested the robustness of SCASL to data sparsity and lack of coverage, i.e., tolerance to forced dropouts of AS probability values or randomly removed genes. As shown in Fig. S1C, even in the cases when 50% of the AS probability values were dropped out, SCASL still largely reproduced the cell clustering landscapes obtained from the original data. Randomly removing large proportions of the AS sites or genes from the original data also did not compromise the cell clustering patterns (Fig. S1D). These results indicate the redundancy of information within the splicing landscapes for defining cell clusters by SCASL.

Finally, to test the stability of the clustering results of SCASL, we performed random sub-sampling of the cells (100%, 75%, or 50%),

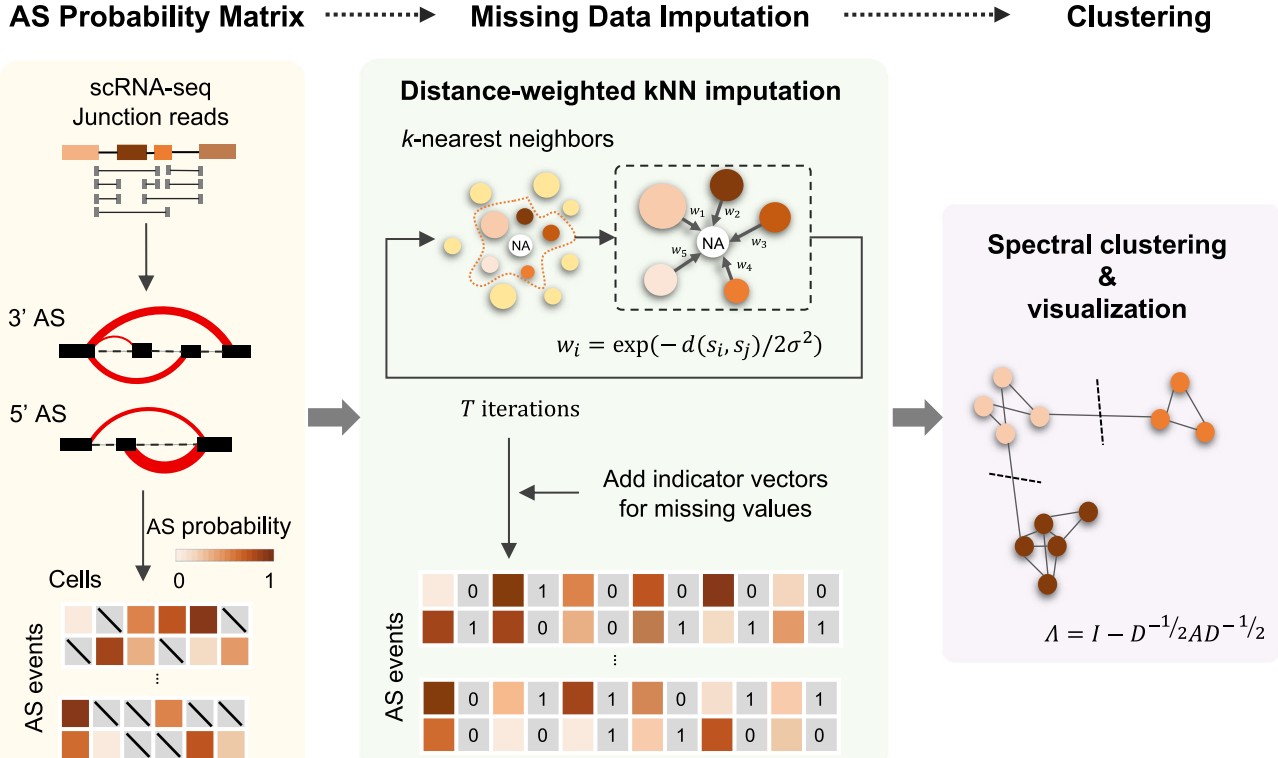

**Fig. 1 | Schematic overview of the SCASL pipeline.** SCASL takes raw scRNA-seq data as input and generates classifications of cell subpopulations. The pipeline is composed of 3 major steps: establishment of an AS probability matrix from the input data, imputation of the missing values in the matrix, and spectral clustering of the single cells.

**Table 1 | Data resources of single-cell RNA-seq**

| Cell source | Species | Method | Cell number | AS events | AS genes | Reference |
|---|---|---|---|---|---|---|
| TNBC | Homo sapiens | Smart-seq2 | 422 | 6111 | 2178 | 33 |
| TNBC | Homo sapiens | Smart-seq2 | 443 | 3565 | 2248 | 34 |
| Hepatoblast | Mus musculus | Smart-seq2 | 486 | 3204 | 1572 | 47 |
| HCC Immune cells | Homo sapiens | Smart-seq2 | 2349 | 10257 | 3879 | 61 |
| Brain | Mus musculus | Smart-seq2 | 1734 | 3058 | 1615 | 55 |
| GBM Immune cells | Homo sapiens | Smart-seq2 | 1218 | 8235 | 2795 | 98 |
| Brain | Mus musculus | 10X | 2442 | 267 | 170 | 97 |

which were then used for cell clustering based on their splicing or gene expression profiles (Figs. S2A, B). For the group of 100% sub-sampling, clustering procedures were simply repeated with the full datasets, and the results provide an assessment of robustness when different random seeds were used by the computer. As shown in Fig. S2C, for different ratios of sub-sampling, the cell clusters defined by SCASL based on single-cell AS profiles are much more robust than the ones defined by Seurat based on the gene expression profiles. When repeated with the full datasets, SCASL also demonstrated greater stability compared to the expression profile-based method.

**SCASL reveals heterogeneous tumor cell clusters, providing new insights into tumor progression**

Due to extensive intra-tumor heterogeneity of single-cell gene expression profiles, it has been technically challenging to define cancer cell clusters with clear physiological relevance[32]. With a dataset of 422 tumor cells (259 primary tumor cells and 163 micrometastatic cells) from one patient with triple-negative breast cancer (TNBC)[33], SCASL defined 3 clusters (Fig. 2A), which were largely inconsistent with the cell clusters defined based on the single-cell gene expression data

(Figs. 2B, S3A). Specifically, one of the newly defined clusters based on AS, C0, was dominated by micrometastatic cells (Figs. 2A, S3B). The genes upregulated in C0 served as strong prognostic markers for patients with micrometastatic breast cancer with worse survival rates (Fig. S3C). The other two cell clusters, C1 and C2, were mainly composed of primary tumor cells, with small proportions of micrometastatic cells (Figs. 2A, S3B).

Among the three clusters of tumor cells, those in C2 were generally more differentiated than the others (Figs. 2C, S4A). Note that the cell clusters defined by gene expression profiles did not show such significant differences in their differentiation stages (Fig. S4B). Furthermore, compared to the other tumor cells, the cells in C2 were characterized by lower expression of genes involved in tumor-promoting processes, such as G2/M phase transition, nuclear division, DNA replication, response to hypoxia, and *Wnt* signaling (Fig. 2D). Therefore, we suspect that this cluster indicates a tumor cell subtype with less malignancy and potentially represents an early intermediate stage during TNBC cell development. We then compared these three clusters of tumor cells with normal breast epithelial cells from another published study of TNBC[34]. Interestingly, despite being doubtlessly

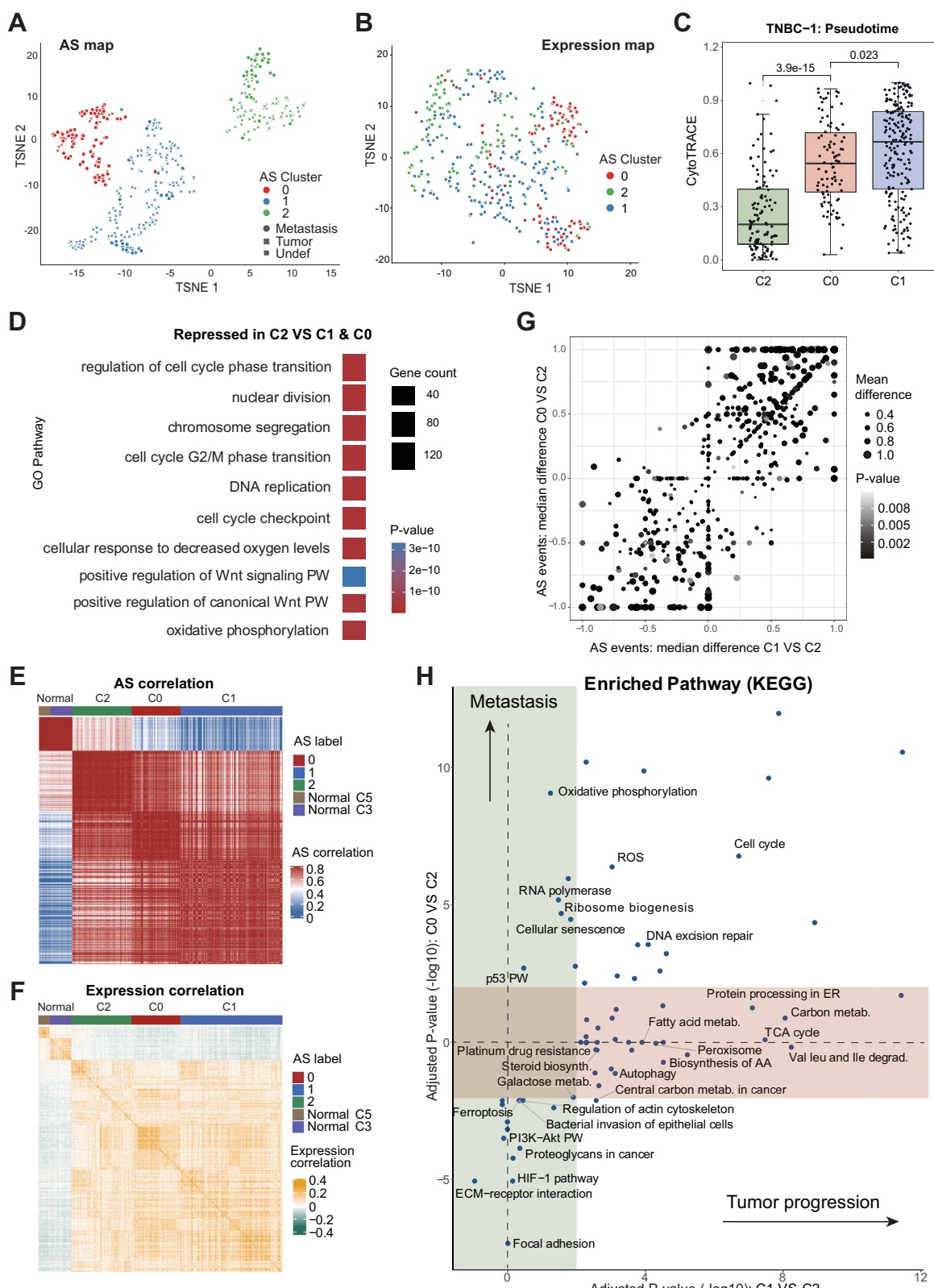

classified as tumor cells, only the C2 cluster showed a great extent of similarity with normal epithelial cells based on AS profiles (Fig. 2E). This suggests that C2 cells may indeed be an intermediate stage during the malignant transformation of normal epithelial cells into cancerous cells in TNBC. Interestingly, such similarity between C2 and normal epithelial cells cannot be seen from the gene expression profiles (Fig. 2F), suggesting that the AS profiles indeed brought in additional

and critical information to define cell identities. In addition, none of the tumor cell clusters defined in the original study based on the gene expression profiles showed such a pattern of similarity and thereby revealed such early-stage tumor cells (Figs. S4C, D).

Furthermore, the analysis of single nucleotide variations (SNVs) with Monovar[35] showed that the cells in C1 bear significantly more SNVs than the cells in C2 (Fig. S4E). This is consistent to our hypothesis

**Fig. 2 | TNBC tumor cell heterogeneity characterized by SCASL. A, B** UMAP plot showing clustering of 422 TNBC tumor cells by SCASL based on the AS landscapes (**A**) or clustering by Seurat based on the gene expression profiles with 3500 variable features. The cells are color labeled by the clusters defined by SCASL. The principal component analysis (PCA) is performed with a dimensionality reduction number of 20 for both methods. Source data are provided as a Source Data file. **C** Pseudotime analysis by CytoTRACE for the three cell clusters (the numbers of cells in C2, C0, and C1 are: 115, 94, and 196 respectively). Wilcoxon rank-sum test were used to evaluate the statistical significance between different groups (two-sided test, using Bonferroni correction to adjust for multiple comparisons). The *X*-axis is sorted from small to large according to the CytoTRACE value. The *p*-values of the differences between C2 and C0, and C0 and C1 are 3.9e-15 and 0.023 respectively. Data are presented as median values +/- SEM. Each box shows the median and inter-quartile range (IQR 25th–75th percentiles). Source data are provided as a Source Data file. **D** Gene Ontology (GO) functional enrichment of the genes downregulated in C2 cells compared to the other tumor cells (Wilcoxon test, two-sided test, *p*-value < 0.01). **E, F** Spearman correlations between the splicing profiles (E) or gene expression profiles (F) of TNBC tumor cells and normal breast epithelial cells. **G** Differential splicing profiles of C1 (*X*-axis) or C0 (*Y*-axis) compared to C2. Dot sizes represent the mean differences in the AS probabilities, and the *p*-values of differential splicing calculated by Fisher's exact test are shown in grayscale (two-sided test). **H** Kyoto Encyclopedia of Genes and Genomes (KEGG) pathway enrichment analysis of the differentially expressed genes in C1 (*X*-axis) or C0 (*Y*-axis) compared to C2, represented by adjusted *p*-values obtained from GSEA.

that the cluster of C2 represents an early stage of tumor cells, whereas C1 represents a more malignant stage. By contrast, the cluster of C0, which mainly consists of the cells from micro-metastatic sites, exhibited slightly elevated rates of mutations, indicating that micro-metastasis takes places during early stage of tumor development. This is consistent to the previous notes[36,37]. It should be noted that due to lack of non-cancerous cells from the same patient, these SNVs were identified by alignment of the RNA-seq reads with the human genome reference. They are not all somatic mutations. Nevertheless, the general trend of SNV numbers in the three clusters supports the proposed cellular patterns of tumor development revealed by SCASL based on the AS landscapes.

Next, by comparing the AS landscapes of C1 and C0 to that of C2, we found that the micrometastatic cell cluster C0 and the primary tumor cell cluster C1 represented two branched trajectories of tumor development, rather than a sequential lineage (Fig. 2G). This observation is in line with the current understanding of the development of tumor micrometastasis at early stages of breast cancer[36,37]. Indeed, compared to C2, C1 and C0 were characterized by different patterns of gene expression profile shifts (Fig. S4F). As a result, in addition to a number of biological processes related to cancer development that were perturbed in both clusters of C1 and C0, many other processes and pathways were upregulated or downregulated only in C1 or C0 (Fig. 2H). Specifically, key metabolic processes, such as the TCA cycle, amino acid metabolism, and steroid biosynthesis, were upregulated only in C1 (Fig. 2H), whereas the cells in C2 were characterized by repression of cell-cell/ECM interactions, ferroptosis, and signaling pathways such as the *HIF-1* and *PI3k-Akt* signaling. Together, the results above indicate that the landscapes of RNA splicing indeed reveal additional insights into heterogeneous tumor cell populations with high physiological relevance.

### SCASL identifies precancerous epithelial cells in TNBC based on AS profiles

With another dataset of TNBC[34], which consisted of 382 primary tumor cells and 61 normal epithelial cells from 6 patients, SCASL first correctly identified malignant and other normal cell types, including lymphocytes, macrophages, epithelial and stromal cells (Fig. S5A). Further clustering of the tumor cells and normal epithelial cells identified two clusters of nonmalignant epithelial cells, C5 and C3, the latter of which appeared to be closer to the malignant cells (Fig. 3A). Although the clustering result based on gene expression data can also classify the tumor and normal cells into different clusters, it did not capture the pattern of C3 being positioned in-between the normal cell cluster C5 and other tumor cell clusters (Fig. S5B). Indeed, compared to C5, C3 showed much higher similarity to other tumor cell clusters based on the single-cell AS profiles (Fig. 3B), a phenomenon not reproduced by the gene expression data of the cells (Fig. 3B). Pseudotime analysis further showed that the epithelial cells in C3 were generally less differentiated than the other normal epithelial cells in C5 but more differentiated than the cells in the malignant cell clusters (Figs. 3C, S5C).

The signature genes of both C5 and C3 included well-known negative markers of epithelial-mesenchymal transition (EMT), such as *PRG4*[38], *EFEMP1*[39] and *IFI16*[40,41]. Interestingly, the expression levels of these negative EMT markers were much lower in C3 than in C5 and were further decreased in the tumor cell clusters (Fig. S5D). On the other hand, signature genes known to promote tumorigenesis, such as *SMARCE1*[42], *SPINT2*[43] and *CPNE1*[44], were upregulated in C3 compared to C5 and, as expected, further upregulated in the tumor cell clusters (Fig. S5D). These data above suggest that C3 represents a potentially intermediate stage during malignant development of TNBC cells from normal epithelial cells. Indeed, as shown in Fig. S5E, the cells in C3 bear significantly more SNVs than the cells in C5. As expected, the tumor cells in C2 and C0 cells from the same patient bear more mutations than C3. Moreover, we used inferCNV[45] to infer the DNA copy number changes for each single-cell, which are summarized as an cumulative CNV score. As shown in Fig. S5F, the cells in C3 bear intermediate levels of DNA copy number changes, which are higher than the normal cells in C5 but lower than the tumor cells in C2 and C0, a pattern highly consistent to that of the SNVs.

The normal epithelial cells in C3 and C5 were obtained mostly from patient PT089 (Fig. S5G), indicating that the potential differences between these two clusters of normal epithelial cells were not due to heterogeneity between patients. The RNA splicing landscapes in C3 and C5 showed significant differences for the genes involved in cancer-related processes such as cell adhesion, apoptosis, *ERBB* signaling, and cadherin binding (Fig. 3D), suggesting that the AS landscape shift in the C3 normal epithelial cells contributed to tumorigenic perturbations of cellular processes.

Finally, we compared the gene expression profiles between C3 and C5 (Fig. 3E). In general, the genes downregulated in C3 were significantly enriched in classical antitumoral pathways, such as those related to ferroptosis, *P53*, *TNF* signaling, and cellular senescence, while the upregulated genes were related to pathways promoting cell localization, survival, and proliferation, such as the negative regulation of apoptosis, ECM-receptor interaction, *PI3K-Akt*, Hippo, and *HIF-1* signaling pathways. Some of these characteristics could be conveniently attributed to AS profile shifts. For example, the *ERBB* signaling pathway, which was perturbed by AS in C3 compared to C5 (Fig. 3D), promotes breast cancer tumorigenesis by activating the *PI3K/Akt* pathway[46], which was indeed upregulated in C3 (Fig. 3E).

Taken together, the results above strongly suggest that although they were classified as noncancerous normal epithelial cells, the C3 cells already contained perturbations of some key cancer-related processes, potentially due to shifts in the AS landscape. Therefore, the RNA splicing landscape, as characterized by SCASL, can identify potential precancerous transitional cellular stages, thereby shedding light on the transformation of normal cells into tumor cells, which has been overlooked by mining of only single-cell gene expression data.

### SCASL reveals cell development lineages and potential crosstalk

In addition to the studies above of tumor cells, we also investigated the splicing landscapes of cells in normal development processes. With a

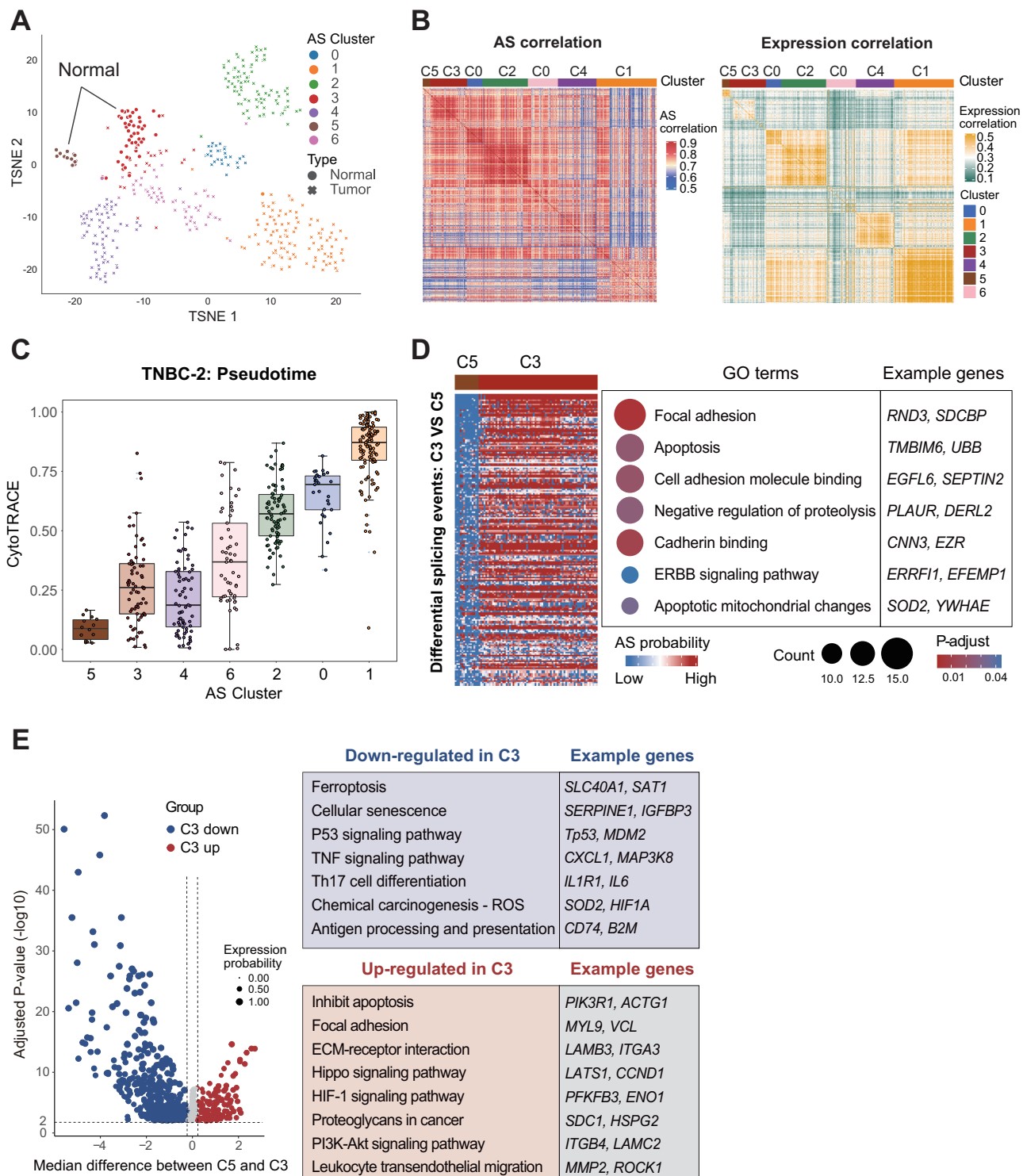

**Fig. 3 | SCASL identifies precancerous epithelial cells in TNBC based on AS landscapes. A** TSNE plot showing clustering of 443 normal epithelial cells and primary TNBC tumor cells based on AS landscapes. The principal component analysis (PCA) is performed with a dimensionality reduction number of 20. Source data are provided as a Source Data file. **B** Heatmaps showing Spearman correlations between all normal and tumor cells based on AS (left) or normalized gene expression (right) profiles. **C** Pseudotime analysis by CytoTRACE for the clusters defined by SCASL (the numbers of cells are 14, 70, 73, 57, 86, 29, and 114, from left to right). Data are presented as median values +/- SEM. Each box shows the median and interquartile range (IQR 25th–75th percentiles). Source data are provided as a Source Data file. **D** The heatmap on the left shows the AS probabilities of the differential splicing events in C3 compared to C5 (p-value < 1e-5, hypergeometric distribution test were used in enrichment analysis, one-sided test). The dot plot on the right shows the pathways enriched by the genes bearing the differential spliced events, and some examples are provided. **E** Volcano plot showing the differentially expressed genes (t-test, two-sided test, adjusted p-value < 0.01) in C3 vs. C5. Differences in the median gene expression levels between the cells of C3 and C5 are shown on the X-axis, and the statistical significance values of the differentially expressed genes (adjusted p-values) are shown on the Y-axis. The sizes of the dots represent the proportions of the cells with expression. Biological functions and processes enriched in the differentially expressed genes are listed to the right, and some representative genes are provided. Source data are provided as a Source Data file.

scRNA-seq dataset of embryonic hepatoblasts[47], SCASL recapitulated two cell lineages in the development of hepatocytes and cholangiocytes from hepatoblasts (Fig. 4A), which have been well described by previous studies[48,49]. Specifically, the orders of the cell clusters were as follows: C2-C10-C3-C1-C8-C5 for hepatocyte development and C0-C9-C6 for cholangiocyte development (Figs. 4A, S6A). Such orders of the cell clusters in these two lineages were confirmed by the embryonic development time (Fig. S6B) and the pseudotime inferred by Cytotrace (Fig. 4B) or Monocle (Fig. S6C). When comparing the gene expression profiles, the two series of hepatocyte and cholangiocyte clusters showed accumulative enrichment of the biological processes essential for the physiological functions of hepatocytes and cholangiocytes, respectively, along the proposed lineages (Fig. 4C, D).

The similarity matrix among the splicing profiles of these cell clusters was well in line with the proposed lineages above (Fig. S6D). Next, we identified the signature AS sites along the two lineages by comparing each cluster to its nearest precursor in the lineage. Together, these sites showed a clear accumulative shift of the splicing landscapes along the proposed multistage differentiation lineage from hepatoblasts to mature hepatocytes and cholangiocytes (Fig. 4E, F). Note that once the orders of the cell clusters in each lineage were shuffled, the same analysis procedure would not produce such patterns of accumulative shifting of the splicing landscapes (Fig. S6E, F).

More interestingly, the single-cell RNA splicing landscape revealed a special cluster, C4, which was composed of annotated cholangiocytes but positioned between the two major developmental lineages for cholangiocytes and hepatocytes (Fig. 4A). Compared to the poorly differentiated cholangiocyte progenitor cell cluster C0, C4 was more differentiated, as shown by their pseudotimes (Fig. 4B). Unexpectedly, although the cells in C4 highly expressed genes enriched in cholangiocyte functions, enrichment of genes in hepatocyte functions such as fatty acid metabolism, steroid metabolism, and alcohol metabolism was also found in C4 (Fig. 4G). Indeed, although the dynamics and molecular mechanisms remain to be revealed, transdifferentiation between cholangiocytes and hepatocytes in both directions has been well described in previous studies[50–54]. Therefore, the fact that C4 possesses the signatures of both cholangiocyte and hepatocyte functions leads to a speculation that this cell cluster represents a potential crosstalk bridging the transition from cholangiocytes to hepatocytes, or vice versa. However, more direct evidence is need to draw a firm conclusion.

Finally, it worth nothing that the two major lineages of hepatocytes and cholangiocytes can be roughly classified based on gene expression data (Fig. S7A), which has been reported in the original study[47]. However, such a cluster distribution pattern does not reflect the time-dependent differentiation trajectory (Fig. S7B). The potential intermediate cell cluster between the lineages of hepatocytes and cholangiocytes is not apparent (Fig. S7C).

As another example, based on the RNA splicing landscapes of mouse brain cells[55], SCASL revealed the well-acknowledged differentiation trajectory from oligodendrocyte precursor cells to oligodendrocytes and astrocytes[56] (Fig. S8A). In addition, microglia and endothelial cells formed two small clusters very close to each other (Fig. S8A). Microglia are derived from blood monocytes/macrophages[57,58], and studies have shown that monocytes can also differentiate into endothelial progenitor cells (EPCs), which migrate and give rise to endothelial cells (ECs) in response to proangiogenic stimuli[59,60]. Some of the ECs in this cluster still expressed the EPC marker CD133 (Fig. S8B). Therefore, the closeness between microglia and ECs may reflect their common ancestors. Such informative patterns were not observed in the clustering results simply based on the single-cell gene expression profiles (Fig. S8B). In summary, by interrogating single-cell splicing landscapes, SCASL can reveal the transitional stages of cell lineages, even for small subpopulations of low-frequency cells.

## Fine clustering of tumor-associated T cells based on RNA splicing heterogeneity

We then used SCASL to delineate heterogeneous tumor-associated lymphocytes based on their RNA splicing landscapes. First, immune cells from different tissues of 6 hepatocellular carcinoma patients[61] were precisely clustered into different major types (Fig. S9A). Next, we looked further into the clusters of T cells (Fig. 5A). The two major types of CD4+ and CD8 + T cells were identified based on either the AS (Figs. 5A, S9B) or gene expression profiles (Fig. S9C). The finer clusters of T cells classified by SCASL based on the RNA splicing landscapes are largely different than those classified by the conventional Seurat pipeline simply based on gene expression data (Figs. S9D, S10).

The tissue distribution preferences of the major subtypes of T cells defined by the AS profiles were consistent with previous studies[61], with CD4+ Treg cells (C3 and C4) and exhausted CD8 + T cells (C6 and C9) being enriched in tumors and circulating CD8 + T cells and central memory CD4 + T cells (C5) being enriched in blood (Fig. 5B). Both C3 and C4 were composed of tumor-infiltrating immunosuppressive regulatory CD4 T cells, as indicated by their high expression of FOXP3[62] (Fig. 5B). However, C4 exhibited stronger immunosuppressive and proliferative features than C3, such as higher expression of the immunosuppressive Treg cell markers CCR8[63,64], LAIR2[65], and BATF[66,67].

On the other hand, although the canonical markers of immunosuppressive Tregs were not highly expressed in C3, C3 Tregs were characterized by almost exclusively high expression of ENTPD1 (CD39) (Fig. 5B). CD39 controls a rate-limiting step for the synthesis of adenosine, which inhibits a series of antitumor processes in the tumor microenvironment by targeting T cells, NK cells, macrophages, and DCs[68,69]. Therefore, C3 represents a special subtype of tumor-infiltrating Tregs with tumor-promoting activities. Such distinctive Treg clusters (C3 and C4) could not be clearly differentiated based on the single-cell gene expression profiles (Figs. S9D, S10), suggesting that the AS landscapes are critical for classifying these differentially functional Treg clusters.

Several genes showed distinct splicing patterns in C3 and C4 Tregs (Figs. 5C, S11A). For example, among the most differentially spliced genes, the SP protein kinase CLK1 (CDC2-like splicing factor kinase) is required for cell cycle progression by regulating the periodic alterative splicing program during the cell cycle[70]. The isoform of CLK1 with skipped exon 4 (CLK$^{T1}$), which was dominant in C3 (Figs. 5C, S11A), has a disrupted catalytic domain of CLK1[71,72], which should result in cell cycle defects and arrest of proliferation via a perturbed cell cycle-related AS program[70,71]. Note that the overall expression levels of CLK1 were not significantly different between the two clusters (Fig. S11B). Therefore, the altered splicing of CLK1 in C3 could potentially contribute to the exhaustion of Tregs. Similarly, several other genes involved in apoptosis and proliferation, including CTSC[73], ANAPC11[74] and RNF213[75], were also subjected to differential splicing, but not differential expression, in the two clusters of Tregs (Figs. 5C, S11B). Further investigation into whether and how these different transcript isoforms function in determining the activation or exhaustion of tumor-infiltrating Tregs is warranted.

## Classifications of tumor-infiltrating active and exhausted CD8 T-cell clusters

Among the CD8+ clusters, C8 from peripheral blood was enriched by antitumor effector T cells (Teffs) marked by genes associated with T-cell activation (CX3CR1 + , FCGR3A + , CCR7-)[76,77] (Fig. 5B). On the other hand, among the 3 tumor-infiltrating CD8 T-cell clusters (C0, C9, C6), both C9 and C6 were marked by signature genes of exhausted T cells (Texs), such as PDCD1 and LAG3[78,79] (Fig. 5B), whereas C0 cells highly expressed the marker of T-cell activation CD160[80], suggesting that C0 was an activated tumor-infiltrating Teff cell cluster.

First, the circulating and tumor-infiltrating active CD8 T-cell clusters (C8 and C0) were marked by differential expression of key

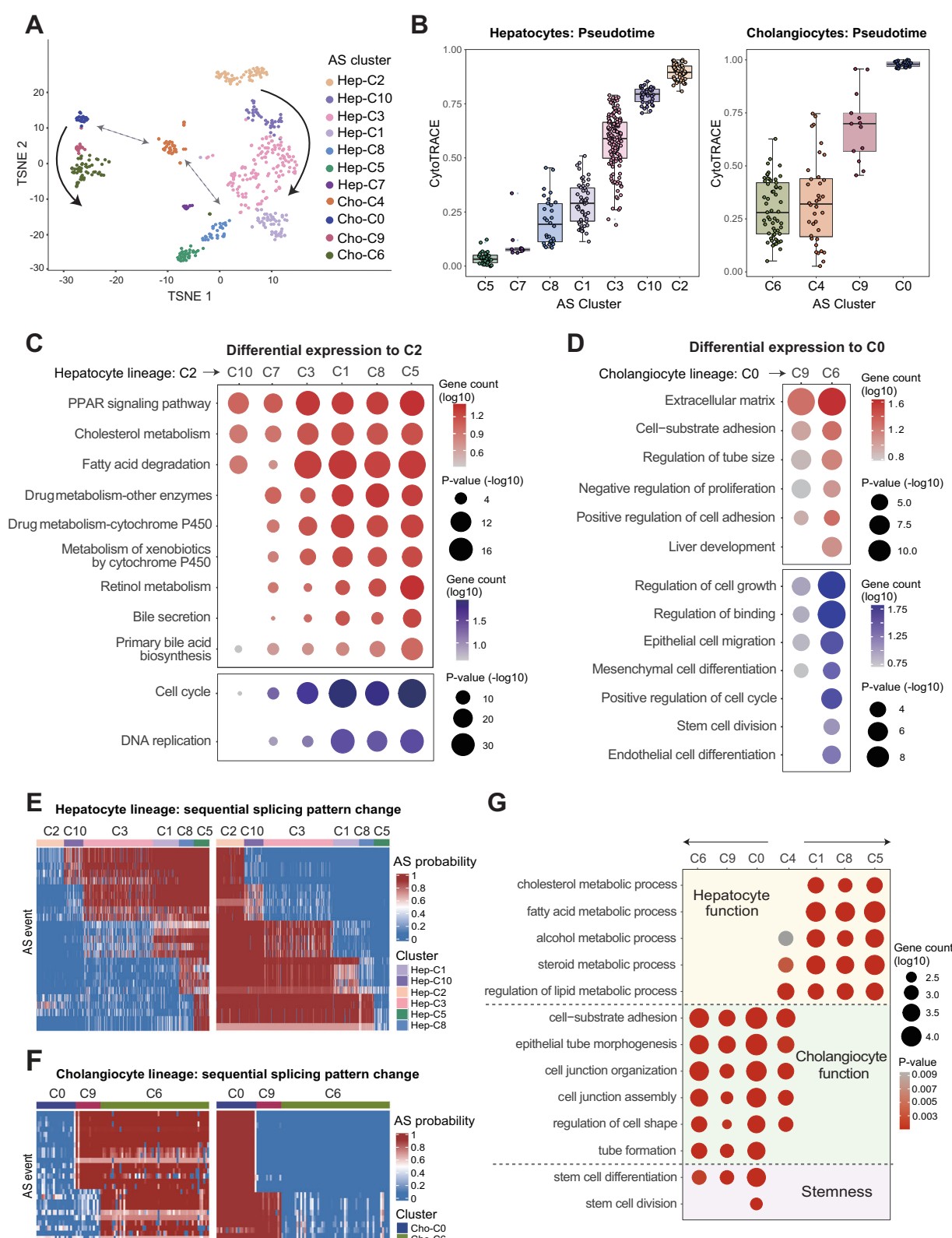

genes related to T-cell infiltration, such as chemokine-related genes (*CCL3L3, CXCR4, CCL3, CCL4*) that are highly expressed in C0 (Fig. 6A), as well as other T cell functions, such as adhesion, cytokine production and responses (Fig. S12A).

The C8 and C0 clusters showed differential splicing profiles for a series of genes, such as *GPR171*, *RSRP1*, and *PTPRC* (Figs. 6B, S12B). *PTPRC* (*CD45*) is critical for T-cell activation because it serves as a

positive regulator of T-cell coactivation upon binding with *DPP4*[81]. Here, the cells in C8 were dominated by the isoform of *PTPRC* with all three exons (4, 5, and 6) included, whereas the cells in C0 express the isoform with all these exons skipped (Figs. 6B, S12B). Previous studies have shown that skipping of these 3 exons, which gives rise to the *CD45RO* isoform, is critical for activated and memory T cells[82,83]. However, more recent studies have shown that activated T cells re-

**Fig. 4 | SCASL identifies developmental lineages of embryonic hepatoblasts.**
**A** TSNE plot showing clustering of 486 embryonic liver cells by SCASL based on AS profiles. The principal component analysis (PCA) is performed with a dimensionality reduction number of 20. The arrows in the figure represent the approximate sequence of embryonic days corresponding to each cluster. Source data are provided as a Source Data file. **B** Pseudotime analysis by CytoTRACE for the hepatoblast/hepatocyte and cholangiocyte clusters. Seven hepatoblast/hepatocyte clusters contained a total of 349 cells (the numbers of cells are 31, 11, 30, 52, 140, 39, and 56, from left to right), and four cholangiocyte clusters contained a total of 137 cells (the number of cells are 57, 36, 13, and 21, from left to right). Data are presented as median values +/- SEM. Each box shows the median and interquartile range (IQR 25th–75th percentiles). Source data are provided as a Source Data file. **C**, **D** Biological function enrichment analysis of the differentially expressed genes (Wilcoxon test, two-sided test, *p*-value < 1e-5) in each cluster of the hepatocyte lineage compared to C2 (**C**) or in each cluster of the cholangiocyte lineage compared to C0. Red indicates upregulation, and blue indicates downregulation. **E**, **F** AS profiles of the top differential splicing events based on pairwise comparisons between the adjacent clusters along the hepatocyte (**E**) and cholangiocyte (**F**) lineages. **G** Biological function enrichment analysis of the differentially expressed genes for each cluster compared to all the other cells (Wilcoxon test, two-sided test, *p*-value < 1e-5). Functional processes related to hepatocytes, cholangiocytes, and stemness were selected and displayed in the dot plots.

express the *CD45RA* isoform with exon 4 included over time, thereby generating more cytotoxic TEMRA cells[84]. Furthermore, new evidence has also indicated that CD8 T cells expressing *CD45RC* (exon 6 included) are more cytotoxic[85,86] and that cells with higher *CD45RB* (exon 5 included) expression have a survival advantage[87,88]. Nevertheless, it is still unclear how these three exons of *PTPRC*, individually or in combination, contribute to the functions of *CD8* T cells. Here, our results suggest that the skipping or inclusion of these three exons marks, and perhaps drives, tumor-infiltration of the activated *CD8* T cells.

Next, we examined the two tumor-infiltrating *CD8* T-cell clusters, i.e., the activated T-cell cluster C0 and the exhausted T-cell cluster C6. The cells in C0 and C6 were originally classified into one cluster based on their gene expression profiles[61]. However, as defined by SCASL based on the AS landscapes, C6 and C0 showed differential expression profiles of many genes related to T-cell activation and exhaustion (Fig. 6C). Specifically, the exhausted T-cell cluster C6 exhibits upregulated apoptosis, downregulated MAPK and NF-kappaB signaling, and increased IFN-gamma signaling, whereas C0, the activated tumor-infiltrating *CD8* T cells, showed the opposite features, i.e., activated proliferation and differentiation signaling, repression of apoptosis, and increased IL-2 signaling (Fig. 6C).

Differential splicing analysis revealed genes with distinct splicing patterns in C6 compared to C0 (Figs. 6D, S13). For example, exon 5 of *HLA-A* and *HLA-C* encodes the transmembrane hydrophobic region of HLA class I molecules, and therefore, exon 5 skipping results in the secretion of HLAs in soluble forms[89–91]. Both *HLA-A* and *HLA-C* were predominantly spliced into soluble isoforms with exon 5 skipped in the exhausted *CD8* T cells of C6, whereas the T cells in C0 expressed insoluble isoforms with exon 5 included (Figs. 6D, S13). Indeed, soluble HLA class I molecules are known to trigger apoptosis of activated *CD8* + T cells[92], providing a potential reason for the exhaustion of the *CD8* T cells in C6.

Another gene with significant splicing differences in C6 vs. C0, *NKG2A*, has been reported to be expressed in NK cells and *CD8* T cells[93] as the canonical isoforms with full length (*NKG2A*) or an alternative functional isoform with exon 5 skipped (*NKG2B*)[94]. C6 was dominated by expression of *NKG2A* (Figs. 6D, S13), which dimerizes with *CD94* to transmit immunosuppressive signals by binding to *HLA-E*[95]. In contrast, C0 mainly expressed *NKG2B* (Figs. 6D, S13), which mainly dimerizes with the an alternative isoform of *CD94* (*CD94-T4*)[96]. Note that CD94 in both C0 and C6 existed as *CD94*-full. Therefore, the isoform switching from *NKG2B* to *NKG2A* in C6 vs. C0 may contribute to the enhanced immunosuppressive signals and therefore the exhaustion of tumor-infiltrating *CD8* T cells.

Many other differentially spliced genes in the exhausted *CD8* T-cell cluster C6 vs the active cluster C0 (Fig. S13) have not been reported before. Given the distinctive splicing patterns of these genes in the two tumor-infiltrating *CD8* T-cell clusters, further studies of whether and how their functions are determined by these AS events and whether their alternative isoforms function differently in the regulation of *CD8* T-cell activation and exhaustion is warranted. Finally, it is worth noting that many of the differential splicing events did not result in differential expression of the genes (Fig. S11B); as such, it is

unfeasible to identify these potentially important genes based on only expression profiles.

## Discussion

Alternative RNA splicing results in the production of different protein products and changes in RNA stability, localization, and interaction with other molecules, thereby acting as a driving force and master regulator of cellular activities and physiological processes. Conceptually, given the following advantages, AS profiles could be superior to gene expression data for the definition of cell identities. (1) AS is a more upstream factor involved in the regulation of cellular activities, whereas gene expression profiles are usually dominated by secondary, downstream responses; (2) AS profiles are largely polarized in single-cell data, i.e., for most AS events, each single cell bears just one dominant form of splicing[9], thereby being largely immune to technical noise; and (3) AS can be quantified as the relative abundance of one isoform in all isoforms, thereby being resistant to gene expression fluctuation due to technical variations and transcription bursts.

In spite of the advantages above, multiple technical challenges have hindered usage of RNA splicing profiles for definitions of cell identities. First, single-cell AS profiles suffer greatly from missing data points due to dropouts during single-cell RNA sequencing analysis and the identification of AS events. Second, coverage of genes and AS events could be very unbalanced across different single cells. Third, many of the canonical clustering methods are unsuitable for processing the largely polarized AS profiles. Finally, with the current common practice of single-cell transcriptome profiling, RNA splicing profiling is much more costly than simple quantification of gene expression levels.

Here, we present a new method, SCASL, for defining cell identities by interrogating single-cell RNA splicing landscapes. Taking aligned RNA-seq files as input, SCASL first performs de novo identification of AS events for each single cell, followed by quantitative assessment of the probabilities for each AS scheme. To address the issue of data sparsity and unbalanced coverage of the single-cell AS profiles, SCASL employs an iterative, weighted KNN process for imputation of the missing values. With forced data drop-out, we have explicitly shown the robustness and fidelity of this process of imputation. Finally, SCASL performs cell clustering with the full AS probability matrix, concatenated with a NA indicator matrix consisting of binary values showing the positions of the originally missing AS probability values in each cell. The NA indicator matrix indicates the data sparsity pattern of the AS matrix before imputation, which should also be informative for the cellular physiological status[3,28]. Therefore, we tested the performance of SCASL without the information of data sparsity, by shuffling the binary missingness values in the NA indicator matrix independently for each cell. As shown in Fig. S14, shuffling the missingness matrix would largely disrupt the cell clusters that we obtained before with SCASL. These new clusters fail to recapitulate important cellular heterogeneities between tumor vs normal cells, primary vs metastatic cells, and different cell development lineages. Therefore, the missingness matrix is indispensable for definitions of cell clusters by SCASL based on AS.

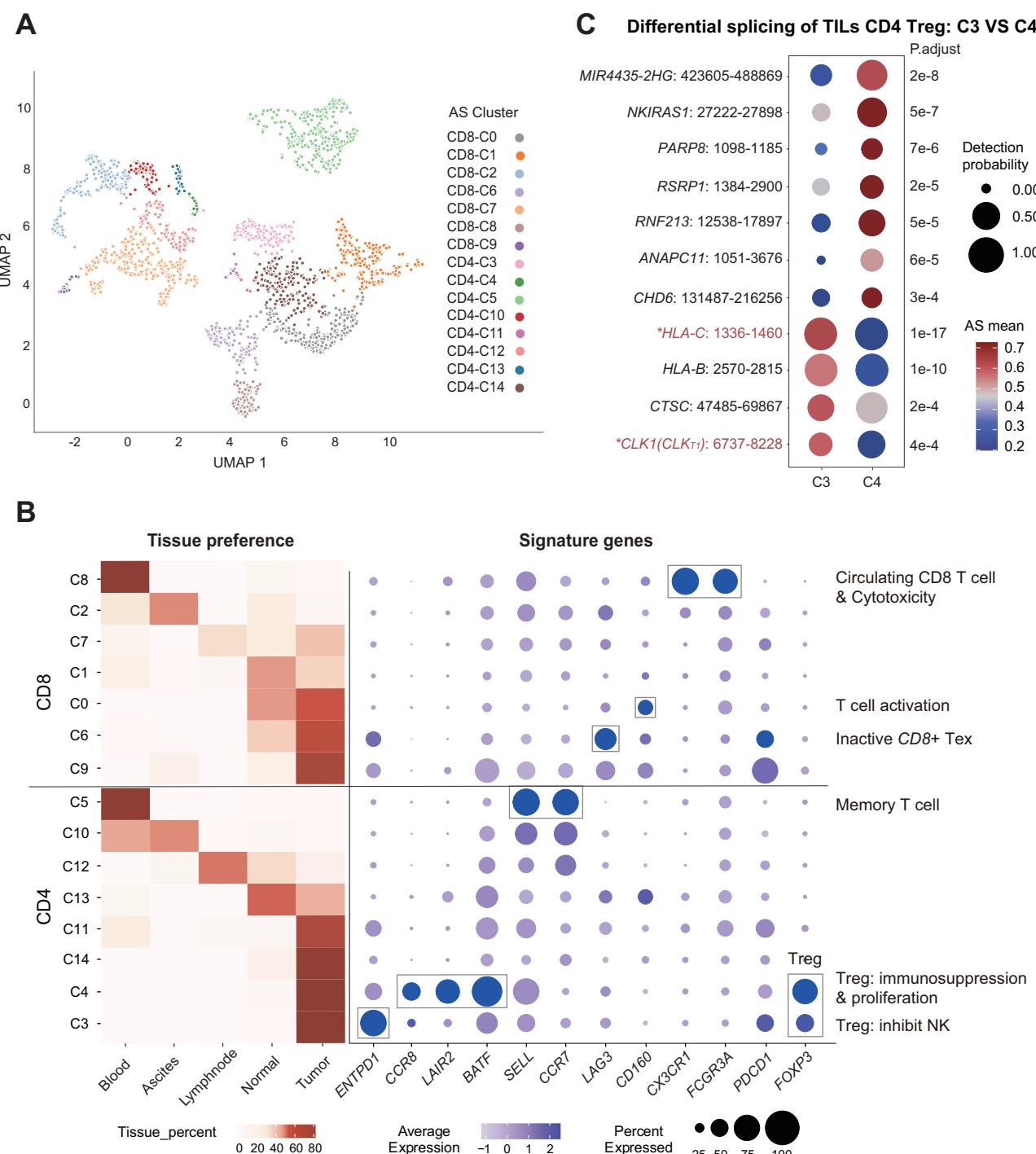

**Fig. 5 | Clusters of tumor-associated T cells defined by SCASL. A** UMAP plot showing clustering of 2349 T cells by SCASL based on AS profiles. The principal component analysis (PCA) is performed with a dimensionality reduction number of 30. Source data are provided as a Source Data file. Source data are provided as a Source Data file. **B** The heatmap on the left represents the distribution preferences of the T-cell clusters in different tissues (calculated based on the number of cells in each tissue/total number of cells in the cluster). The dot plot on the right shows the expression of signature and key functional genes in each cluster. The dot sizes represent the proportions of cells with expression, and the color scale represents

the mean expression level. **C** Top differential splicing events between C3 and C4 Tregs. For each AS event, the dot size represents the proportion of cells in which the splicing probability was detectable from the RNA-seq reads, whereas the color scale represents the average AS probability in these cells. Previously studied AS events are marked in red, while events that have been studied for related functions are additionally marked with an asterisk. *p*-values are shown on the right side of each row (Wilcoxon test, two-sided test). Source data are provided as a Source Data file.

On the other hand, the NA indicator matrix alone is certainly not informative enough to define the cell clusters. The percentages of missing AS probability values (values of NA) are comparable across the cell clusters defined by SCASL (Fig. S15). Furthermore, upon shuffling of

the AS probability matrix, SCASL also failed to recover the cell clusters with biological relevance (Fig. S14). Therefore, a critical and necessary feature of SCASL is to incorporate both the AS profile and the data missingness information for defining cell clusters. In other words, both

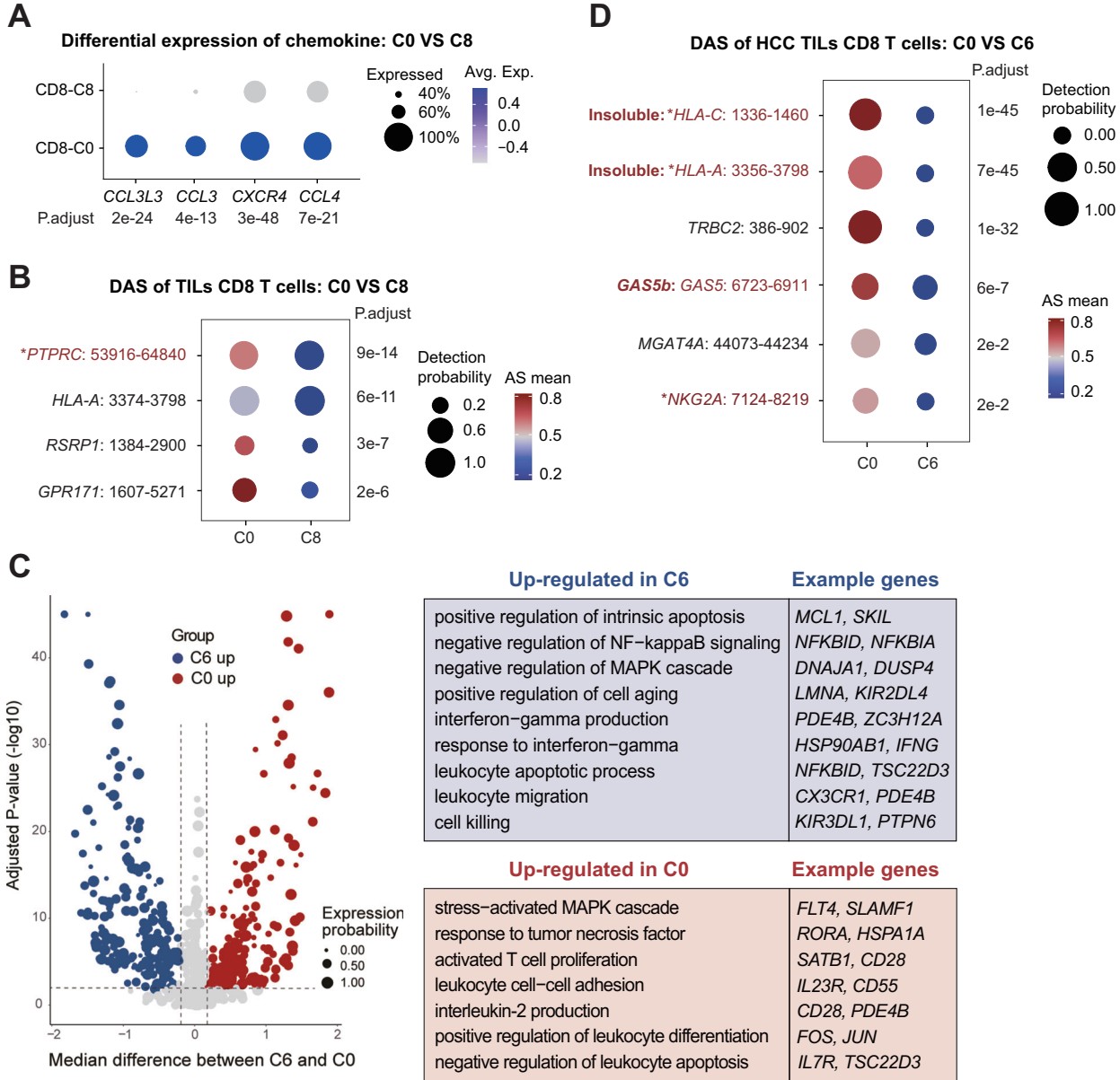

**Fig. 6 | Comparison of the CD8 T-cell clusters defined by SCASL. A** Dot plot showing chemokines with differential expression between C0 and C8. The dot sizes represent the proportions of cells with expression, and the color scale represents the mean expression level. *p*-values are shown on the bottom of each column (Wilcoxon test, two-sided test). **B** Top differential splicing events between the *CD8* T cells in C0 and C8. For each AS event, the dot size represents the proportion of cells in which the splicing probability was detectable from the RNA-seq reads, whereas the color scale represents the average AS probability in these cells. Previously studied AS events are marked in red, while events that have been studied for related functions are additionally marked with an asterisk. *p*-values are shown on the right side of each row (Wilcoxon test, two-sided test). Source data are provided as a Source Data file. **C** Volcano plot showing the differentially expressed genes between C0 and C6 (*t*-test adjusted *p*-value < 0.01, two-sided test). The median

difference in gene expression levels between C0 and C6 cells is shown on the *X*-axis, and the statistical significance values of the differentially expressed genes (adjusted *p*-values) are shown on the *Y*-axis. The sizes of the dots represent the proportions of cells with expression. Biological functions and processes enriched in the differentially expressed genes are listed to the right, and some of the representative genes are provided. Source data are provided as a Source Data file. **D** Top differential splicing events between the *CD8* T cells in C0 and C6. For each AS event, the dot size represents the proportion of cells in which the splicing probability was detectable from the RNA-seq reads, whereas the color scale represents the average AS probability in these cells. Previously studied AS events are marked in red, while events that have been studied for related functions are additionally marked with an asterisk. *p*-values are shown on the right side of each row (Wilcoxon test, two-sided test). Source data are provided as a Source Data file.

the information of AS probabilities and data sparsity pattern contributes to the cell heterogeneity landscapes as portrayed by SCASL.

Profiling of single-cell RNA splicing requires long RNA-seq reads with full-transcript coverage, which are usually obtained with SMART-seq techniques. Compared to droplet-based scRNA-seq methods, which only profile RNA expression levels, SMART-seq and similar methods are much more costly and difficult to multiplex for large cell numbers. This indeed limits the classification of cell identities based on

splicing landscapes in practice. However, as shown in Figs. 3–5 and S5, SCASL recovered minor clusters of infrequent cells, suggesting that it is feasible to define cell subtypes with a relatively smaller number of single-cell samples. Nevertheless, it is well expected that future technical advances would greatly reduce the cost and increase the throughput of profiling single-cell RNA splicing landscapes, thereby allowing more extensive analysis of RNA splicing profiles in complex physiological and pathological contexts.

Nevertheless, we tested the performance of SCASL on a dataset generated by 10x Genomics platform for dentate gyrus cells (in mice at postnatal days 12, 16, 24, and 35)[97]. It is known that neuroblasts differentiate into granule cells. Clustering of neuroblasts and granule cells can be challenging, as both cell types undergo developmental transitions and share certain molecular markers. The cell clusters defined by SCASL based on the highly 3′ biased 10x Genomics single-cell RNA-seq data indeed show transitional differentiation patterns during granule cell development, as shown by differentiation marker genes (Figs. S16A, B). By contrast, such transitions were not recapitulated by the cell clustered defined by their gene expression profiles, which only shows two major groups of neuroblasts and granule cells (Fig. S16C). It should be noted that the cell type annotations were defined based on gene expression profiles in the original study. Therefore, it is well expected to see the two types of cells clearly separated in Fig. S16C.

This analysis shows that despite the strong 3′ bias, 10X RNA-seq data can also be used for cell clustering by SCASL. As discussed previously in the manuscript, the information of cell identities supplied by AS profiles are highly redundant, and therefore, fractions of the AS profiles within the 3′ regions of mRNAs could be informative for definitions of cell clusters. Technically, the procedure of SCASL is exactly the same with either SMART-seq or 10X data. However, it should be noted that most of the 10X RNA-seq libraries were subjected to very low depth of sequencing. Therefore, splicing-based clustering with SCASL is not recommended for the droplet-based 3′ end RNA-seq data with low sequencing depth.

In summary, we propose the SCASL method for cell subtype classification based on systematic interrogation of single-cell RNA splicing landscapes. Given the higher position of RNA splicing in the gene regulation hierarchy, this new scheme for identifying cell identities recapitulates the intrinsic generation of cell heterogeneity from a unique perspective, thereby serving as a tool for revealing novel insights into key cellular processes in diverse physiological contexts. Finally, it should be noted that SCASL has been designed and optimized for processing sparse and polarized single-cell RNA splicing profiles. In its current form, SCASL is not suitable for clustering analysis with bulk RNA-seq data.

## Methods

### Data source and preprocessing

Seven sets of single-cell RNA-seq data generated by the SMART-seq platform and 10X platform were downloaded from the NCBI SRA Database (Table 1). These datasets were previously published for studies of TNBC[33,34], tumor-associated immune cells[61,98], mouse embryonic liver[47] and brain[55,97]. The raw reads were mapped to the reference genomes (human hg19 or hg38, according to the original studies, or mouse mm10) with hisat2, generating BAM files, which would be used as inputs of SCASL. Other read mapping algorithms would work as well, but we recommend hisat2 for its faster speed and higher accuracy especially for identifying the junction reads[99].

### The SCASL pipeline

**Quantification of AS events.** The R package LeafCutter was used to extract and count junction reads. 5′ and 3′ splice sites (5′ SSs and 3′ SSs) were then identified from these AS events. Next, inspired by the strategy used in FRASER[23], the AS events sharing the same 5′ SSs or 3′ SSs were grouped together and defined as an AS module. For each AS event, the junction read count was divided by the count of all the junction reads from all the AS events within the module. This strategy does not rely on transcriptome annotation, and its results provide a comprehensive survey of different splicing patterns, including the canonical AS events (such as exon skipping, intron retention, mutually exclusive exons) as well as other splicing errors due to rare mutations or aberrant spliceosome.

Recent works have highlighted potential limitations of quantifying AS with low-depth 2nd generation sequencing data[100–102]. Studies have also demonstrated that errors stemming from the limitations of sequencing depth can be mitigated by implementing a filtering threshold for splice junction reads of >10[103]. Therefore, here the AS modules with ultralow sequencing depth (no >20 junction reads combined for all the AS sites within the module) were discarded. For each AS event within an AS module, its probability of occurrence was then quantified as the number of junction reads covering the particular AS event divided by the total number of reads for all the AS events within the AS module:

A junction read data entry of value $R_{m,n,l}$ with the m-$th$ 5′-end, the n-$th$ 3′-end and the l-$th$ cell sample can be used to calculate the AS probability as follows:

$$P^{\text{upstream}}_{m,n,l} = \frac{R_{m,n,l}}{\sum_i R_{i,n,l}} \quad (1)$$

$$P^{\text{downstream}}_{m,n,l} = \frac{R_{m,n,l}}{\sum_j R_{m,j,l}} \quad (2)$$

Finally, the AS probability values of all the AS sites in all the cells are arranged into an AS probability matrix, of which each row represents an AS site and each column represents a single cell. Quality filters are then implemented to remove the cells and AS sites with too many dropouts so that eventually, each of the AS sites (i.e., rows of the matrix) has AS probability values in at least 10 single-cell samples, and at least 1000 AS sites have AS probability values in each single cell (i.e., a column of the matrix). Note that these criteria are adjustable parameters of the SCASL algorithm. We also provide a density map of cells and junction sites as a reference to help users select the optimal parameters for datasets with different scales, coverages, qualities, and sequencing depths.

**Imputation of missing data values.** The limited sequencing depth of scRNA-seq poses a challenge in terms of data availability, as a significant proportion (over 98%) of the data entries are not applicable (NA). This data sparsity is a result of biological heterogeneity as well as technical noise. To address this issue, we introduced two solutions. First, to make use of the positional information of missing values, a missing-value indicator was introduced for each data entry. For the m-$th$ site and n-$th$ sample, suppose the original AS probability is denoted as $X_{mn}$, and the missing-value indicator is defined as follows:

$$R_{mn} = 1(X_{mn} = \varnothing) \quad (3)$$

Additionally, we used a novel imputation method to minimize the impact of missing data. The first step is a simple mean imputation. The missing values are estimated by the average of the available values at the same splicing site. Specifically, for the i-$th$ site and the j-$th$ sample, the initial imputed value is estimated to be

$$Z^{(0)}_{mn} = \hat{E}_j\left[X_{mj}\right] \quad (4)$$

Following the coarse mean imputation, a refined imputation is achieved by an iterative weighted KNN imputation. To perform this imputation, the k-nearest neighboring samples are found with the Euclidean distancing metric. The missing values are then imputed with the weighted average of the corresponding AS probabilities from these k neighboring samples which is a common practice in most, if not all, tools based on KNN imputation, e.g., missForest[104], VIM[105], etc. Mathematically, suppose we have a missing data entry $X_{mn}$, and we have already estimated it to be $Z^{(t-1)}_{mn}$; after the first $t-1$ round of imputation,

the imputed value for the t-*th* iteration can be calculated as follows:

$$Z_{mn}^{(t-1)} = \frac{1}{k} \sum_{i=1}^{k} w_i' Z_{mn_i}^{(t-1)} \tag{5}$$

where $n_i$ is the i-*th* nearest neighboring sample, and

$$w_i = e^{\frac{-d(n,n_i)}{2\sigma^2}} \tag{6}$$

$$w_i' = \frac{w_i}{\sum_{i=1}^{k} w_i} \tag{7}$$

is the normalized weight factor related to the Euclidean distance d $(n, n_i)$, $\sigma^2$ the variance of the distances to the k nearest neighbors. This method takes into account the variability of the distances, which can provide insights into the density and distribution of the neighbors.

Determinations of other iteration parameters, i.e., the number of neighbors k and the number of iterations t, requires comprehensive consideration of sample size, data sparsity, and computation costs. Based on our analysis with multiple datasets, the imputation tends to converge after three iterations (i.e., $t = 3$). The selection of the parameter k primarily depends on the size of the dataset. According to the rule of thumb, k is typically set as the square root of the total number of samples or a smaller fixed value[106]. Large k values result in imputations based on too many neighbor cells, which would lead to underestimations of cellular heterogeneity, whereas small k values tend to introduce stochastic noise due to lack of enough reference for imputation. Additionally, the choice of k has a significant impact on the computation cost for large datasets. A smaller k value can greatly improve the imputation speed. After systematic comparisons, we have set the default value of k to 10, which is suitable for a wide range of data with sample numbers ranging from hundreds to thousands. According to our tests with >10 different published datasets, the default parameters ($k = 10$ and $t = 3$) work quite well in data imputation and cell clustering. However, in special cases such as extremely large datasets or abnormally high sparsity, users can easily adjust the parameters to compare the outcomes and get the optimal results.

Finally, the imputed matrix $Z_{mn} = Z_{mn}^{(t)}$ is obtained. By concatenating it with the missing value indicator matrix $R_{mn}$, a more comprehensive data presentation for clustering is derived. Therefore, the final data matrix D is of shape $2M \times N$, where M and N are the number of AS sites and cells, respectively.

**Cell clustering.** We first use principal component analysis (PCA) on data matrix D to perform dimensionality reduction. For the clustering algorithm, we chose spectral clustering considering its low computation cost and stable clustering quality. We also considered its capability of scaling to different data distributions.

Spectral clustering is a graph-based clustering algorithm. First, we construct an undirected graph $G = (V, E)$, where the vertical set $V$ is the set of sample cells and $E$ is the set of weighted edges among the vertices. The weight $w_{ij}$ is supposed to represent the affinity between the pair of cells $v_i$ and $v_j$. In this way, we compute the weights with a calculation similar to Formula 6:

$$w_{ij} = e^{\frac{-2d(v_i, v_j)}{\sigma^2}} \tag{8}$$

All these weights compose the weight matrix $W$. By polarization with the $k$ nearest-neighbor ($k$NN) method, we can acquire the adjacency matrix

$$A_{ij} = A_{ji} = 1\left(v_j \in k\text{NN}(v_i)\right) + 1\left(v_i \in k\text{NN}(v_j)\right) \tag{9}$$

With the adjacency matrix and diagonal degree matrix $D_{ij} = \sum_j W_{ij}$, we can compute the normalized graph Laplacian by the following formula:

$$L = I - D^{-\frac{1}{2}} A D^{-\frac{1}{2}} \tag{10}$$

We then calculate the first $k$ eigenvectors (the eigenvectors corresponding to the $k$ smallest eigenvalues of $L$), and the matrix is considered formed by the first $k$ eigenvectors; the i-*th* row defines the features for cell sample $v_i$. Finally, we cluster the samples in this low-dimensional space with the $k$-means clustering algorithm based on their proximity (squared Euclidean distances) and acquire the desired clustering label $c_i$.

To visualize the result, we further project the PCA-reduced data matrix onto a 2-dimensional manifold with UMAP and create a 2D plot colorized by the clustering labels.

## Benchmark for performances of SCASL

To assess the accuracy of the imputation, we randomly set 10%, 30%, and 90% of the available values as the ground truth and transform them into NA values. Then, we perform imputation on the modified matrix using the SCASL pipeline. Finally, we compare the difference between these imputed values and the ground truth and visualize the distribution of the deviation. Additionally, we conduct a *t*-test to compare the differences in imputation accuracy between different proportions of the manually nullified values (Fig. S1A).

To quantitatively evaluate the difference between expression profile clustering results and AS clustering results, we employ normalized entropy, a metric used to evaluate the quality of clustering results. Given a set of true class labels as ground truth, normalized entropy measures how closely the clustering results match the true classes. Mathematically, suppose there are $l$ categories for the ground truth, and the AS clustering generates $k$ classes. Then, for clustering class $q$, the entropy

$$H_q = -\sum_{i=1}^{l} n_q^j \log \frac{n_q^j}{n_q} \tag{11}$$

measures the disorder of the distribution of the clustering results within category $q$, where $n_q$ represents the number of samples in class $q$, and $n_q^j$ represents the number of samples in class $q$ that is categorized as $j$ by the ground truth. A lower entropy value indicates a higher level of purity in class $q$ and hence a better clustering result. Subsequently, the overall clustering performance is evaluated through a weighted summation of the entropy, where the weights are determined by the proportion of samples in each clustering class ($n$ is the total number of samples):

$$H = \sum_{q=1}^{k} \frac{n_q}{n} H_q \tag{12}$$

To account for the impact of varying numbers of ground truth categories on entropy values, we normalize the entropy by dividing it by the theoretical maximum entropy, which is achieved under a uniform distribution, meaning the clustering result is completely random:

$$H_{\max} = -\sum_{q=1}^{k} \frac{1}{k} \sum_{j=1}^{l} \frac{1}{l} \log \frac{1}{l} = \log l \tag{13}$$

Finally, the normalized entropy can be obtained as follows:

$$\widetilde{H} = \frac{1}{H_{\max}} H = -\frac{1}{n \log l} \sum_{q=1}^{k} \sum_{j=1}^{l} n_q^j \log \frac{n_q^j}{n_q} \tag{14}$$

To assess the robustness of the SCASL method, we not only calculated the normalized entropy of ground truth and SCASL labels but also randomly scrambled 25% of labels in the ground truth (or the expression clustering result) and calculated the resulting normalized entropy between the SCASL labels and the scrambled ground truth labels to compare the two methods. This process was repeated 10000 times, and we visualized the distribution of the normalized entropy for each dataset (Fig. S1B).

Additionally, to test the tolerance of SCASL for missing values, we randomly set 10% of the measured values in each dataset to be NA values and gradually increased the percentage by 10%. We then compared the differences between new clustering results and unmodified clustering results by normalized entropy analysis (Fig. S1C). We also evaluated the redundancy of the AS information by randomly deleting a portion of the junction group in each dataset with a 10% gradient increase in percentage. The differences in the clustering results with and without the junction group deletion were then compared through normalized entropy analysis as well (Fig. S1D).

### Differential alternative splicing analysis

We used Fisher's exact test with screened high-quality alternative splicing events for differential splicing analysis. Specifically, due to the overdispersed single-cell RNA-seq reads, we did not use the read counts directly for differential splicing analysis between different cell groups. Instead, considering the highly polarized AS probabilities in single cells, we turned the differential splicing analysis into a classification problem. For each AS event, we asked whether its occurrence is preferentially enriched in one cell cluster but not in another cluster. Here, an AS event with probability >0.6 is categorized as present in the cell, whereas an event with probability <0.4 is categorize as absent. In fact, the probabilities of most AS events classified as absent were close to 0, whereas the ones classified as present have probabilities close to 1. After such categorizations, we then performed the Fisher's Exact Test for pair-wise comparisons between cell clusters to evaluate whether an AS event is preferentially enriched in one cell cluster but depleted in the other one.

We performed differential splicing analysis of the clusters in each dataset and kept the events with $p$-values >0.01 as differentially spliced events. We then selected the top upregulated and downregulated events for heatmap display (heatmaps were plotted using the ComplexHeatmap package version 2.2.0). Finally, we mapped each event back to the gene with the highest inclusion and the closest distance through the positional relationship between the junction and the gene on the chromosome.

For the potential functionally differential splicing events in the T cells obtained by the above method, we further calculated the splicing mean difference, median difference and detection probability of these events using the available values from unimputed raw data and displayed the top ones in a dot plot, again verifying the accuracy of the differential splicing analysis of the imputed matrix.

### Bioinformatic analysis with single-cell gene expression profiles

We downloaded the processed expression profile data provided in the original paper of each set of data. First, we used Seurat (package version 3.2.2)[31], a traditional single-cell expression profile clustering tool, for quality filtering, normalization and clustering of expression profiles based on the common parameters recommended by the official website of Seurat (Filtering parameters are usually defined as min.cells = 10, min.genes = 1000. For datasets with large differences in cell numbers, 5% of cell numbers and gene numbers are selected as filtering criteria according to the specific circumstances of the data, as was done for the T-cell dataset). Subsequently, the expression profile clustering results were compared with the clustering results of SCASL by dimensionality reduction visualization (UMAP/TSNE) and calculation of normalized entropy. All features in the expression profile and AS profile are used when calculating correlation relationship between cells. Additionally, we selected a set of important genes and visualized their expression ratios and changes in expression levels across each cluster.

We employed inferCNV[45] for copy number variation analysis and MonoVar[35] for single nucleotide variation analysis. We selected events with reads >10 and mutation frequency >0.6 as SNV events. The observations derived from inferCNV were categorized into five distinct groups based on predefined thresholds: complete loss (2 points), loss of one copy (1 point), neutral (0 points), gain of one copy (1 point), gain of two or more copies (2 points). Subsequently, we computed the cumulative CNV score for each individual cell by summing these assigned scores.

### Differential expression analysis

For the signature genes and functional definition of each cluster, in addition to the label that has been annotated in the original study of the data source, We use Seurat's 'Find marker' function to find differentially expressed genes with adjusted $p$-value <0.01 (by default, Wilcoxon test is used for difference comparison, and Bonferroni multiple test is used for $p$-value correction); The selection of an appropriate statistical test for differentially expressed gene analysis in cell clusters depends on the characteristics of the data set, including its distribution and cell number. When special classification comparisons are required, the Wilcoxon test or $t$-test may be employed (FDR is used to adjust the $p$-value), depending on the specific nature of the data.

### Functional enrichment analysis

We conducted functional enrichment analysis on the identified differentially expressed genes and differentially spliced genes to provide initial insights into the functional changes associated with each cluster and key developmental stages. The clusterProfiler package version 3.14.3 was used for gene ontology (GO) and Kyoto Encyclopedia of Genes and Genomes (KEGG) pathway enrichment analysis, and org.Hs.eg.db and org.Mm.eg.db package version 3.10.0 were used for Entrez ID conversion.

### Survival analysis

To further validate the significance of selected important genes, we performed survival analysis using tumor clinical information obtained from the Cancer Genome Atlas (TCGA) based on GEPIA2 platform[107]. We selected the median as the group cutoff and utilized a 95% confidence interval to calculate the 5-year Overall Survival and determine the hazards ratio based on a Cox Proportional Hazards Model.

### Pseudo-time analysis

We utilized gene count data from each dataset to perform pseudo-time analysis on each cell cluster through two widely used trajectory analysis tools, Monocle[108] package version 2.14.0 and CytoTRACE[28] package version 0.2.1, which employ distinct analytical principles. The results from the two methods were compared to verify the robustness of the findings. The accurate differentiation stage relative to each cluster was ultimately defined by integrating the results of the functional analysis for each cluster.

### Statistics and reproducibility

The statistical tests have been described in the sections above and in the figure legends. No statistical method was used to predetermine sample size. No data were excluded from the analyses. The experiments were not randomized. The Investigators were not blinded to allocation during experiments and outcome assessment.

### Reporting summary

Further information on research design is available in the Nature Portfolio Reporting Summary linked to this article.

## Data availability

All single-cell data utilized in this manuscript were sourced from publicly available databases, with the exception of the T cell dataset, which was deposited in the Gene Expression Omnibus (GEO). The accession numbers associated with the datasets are as follows: "GSE123837", "GSE118390", "GSE90047", "GSE71585", "GSE84465" and "GSE95315". The T cell data are available under restricted access for download restrictions on the China National Center for Bioinformation (CNCB) platform, access can be obtained by submitting an application to the Zhang Lab (please contact: hengsj@mail.cbi.pku.edu.cn), and can subsequently be downloaded from the national genomics data center with the accession number "HRA000069". Source data are provided with this paper.

## Code availability

Codes for all the data analyses in this study are available in GitHub (https://github.com/xryanglab/SCASL)[109], https://doi.org/10.5281/zenodo.10678937.

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

## Acknowledgements

We thank the supports from the Tsinghua University Branch of China National Center for Protein Sciences (Beijing) and Tsinghua University Technology Center for Protein Research, including the core facilities of Biocomputing, Genome Sequencing and Analysis at Tsinghua University. This work was funded by the National key research and development program (2023YFC3043300), the National Natural Science Foundation of China (32330022, 81972912, and 31671381), the National Special Support Program for High-Level Talents, the Tsinghua University Spring Breeze Fund, and the Tsinghua University Initiative Scientific Research Program (20221080084).

## Author contributions

Conceptualization, X.X. and X.Y.; Bioinformatics pipeline, X.X.; Data collection and processing, X.X., Y.H., and Z.Z.; Bioinformatics analysis, X.X.; Writing-Original Draft, X.X. and X.Y.; Writing-Review, Modification, and Editing, X.X. and X.Y.; the sequencing raw data of T cells was provided by Y.H. and Z.Z.; Supervision, X.Y.

## Competing interests

The authors declare no competing interests.
