## [Peer Review File · Nature Communications]

Interrogations of single-cell RNA splicing landscapes with SCASL define new cell identities with physiological relevanceReviewer #1 (Remarks to the Author):

The authors develop a pipeline "SCASL" to cluster cells based on alternative splicing (AS), rather than total gene expression. Their pipeline is roughly LeafCutter -> KNN smoothing -> spectral clustering. They show various examples in Smart-seq data where it seems SCASL is pulling out biologically meaningful clusters.

The paper is reasonably clearly written and the methods are reasonable overall. I have two main concerns with the paper however.

First is that for a paper presenting a novel method there is a lack of systematic evaluation. The biological findings are nice but don't really address whether 1) SCASL is really finding more robust or meaningful clusters than gene expression and 2) whether the various choices being made in the pipeline are optimal (e.g. how is sigma chosen in KNN? Does the iterative part of KNN actually improve performance). For (1), it is of course difficult to assess clustering quality. But I think something like (repeatedly) randomly splitting a dataset in two (two sets of cells) and then assessing the reproducibility of the clusters found using SCASL vs Seurat (applied to each subdataset separately) would be a good start.

Second is the addition of the missingness matrix. This sounds perfectly reasonable, but it is very possible that it acts as a proxy for expression. Junctions that are missing in a sample are indicative of low expression of the gene, and vice versa. Thus this missingness matrix is effectively binarized total expression. The smoothing and clustering are therefore getting information about BOTH AS and expression. That's not necessarily a bad thing; indeed to get the best possible cell type resolution this might be very sensible. This needs exploring: can SCASL still cluster at all without the missingness matrix? If not that's OK, but the description of the method should acknowledge this potential issue.

More minor issues:

SCASL uses junction usage relative to other junctions that share either the same 3' or 5' splice site (so each junction will actually be represented twice). This is different from the LeafCutter approach of forming junction "clusters". FRASER2 also introduced a third option which is to compare each junction to the union of all junctions with shared splice sites. It's not clear to me which approach is better, and it's possible different approaches would be better in different contexts, e.g. I believe the FRASER authors argue their approach is particularly appropriate for missplicing caused by rare variants/mutations.

Using AS to discover additional heterogeneity beyond that represented by expression alone has explored before (e.g. <https://elifesciences.org/articles/70692> and <https://academic.oup.com/nar/article/51/5/e29/6985826>, which are not cited/compared to).

It would be interesting to know if SCASL can do anything with 10x data despite it's substantial 3' bias.

I'm a bit concerned about the FET for differential splicing. RNA-seq read counts (including those from UMI-less Smart-seq) are typically overdispersed (in other words each read is not independent), which FET cannot account for. Something like a beta-binomial GLM would be a better alternative.

Spectrum -> spectral clustering

Fig 2c: do you see this relationship if you cluster by expression?

For the TNBC I imagine you could detect somatic mutations from the Smart-seq data: it would be interested if they correlated with your clusters. Maybe beyond of the scope of the current paper however.

Your representation is not "4 dimensional". It's 2D. It's also weird to consider the matrix as 4NxM rather than Nx4M.

Reviewer #2 (Remarks to the Author):

Review Summary

In this manuscript entitled "Interrogations of single-cell RNA splicing landscapes with SCASL define new cell identities with physiological relevance", the authors developed SCASL, a new approach for single-cell clustering based on alternative splicing (AS) information. The pipeline relies on the LeafCutter software to identify AS events. SCASL identifies the AS sharing 5' and 3' splice sites and quantifies AS probability values. Due to high dropouts in single-cell data, the author developed the imputation approach for missing values and applied the spectral clustering approach for clustering analysis.

Mainly described in the manuscript, the authors applied SCASL to analyze publicly available single-cell datasets generated from the SMART-Seq protocol and demonstrate the method's capability. The manuscript shows, in some of the datasets, a better single-cell clustering for identifying cell identities using AS than standard gene expression.

The proposed method generally represents an important step forward using the AS information as another layer for identifying cell identity. However, there are significant concerns (see below). In particular, it is not clear to the reviewer whom this manuscript is written for, a biologist or a bioinformatician/computational biologist.

From the computational perspective, the manuscript lacks details on the new algorithms and benchmarking, particularly for comparing with existing methods/tools. The authors developed the AS probability matrix built on LeafCutter's results, implemented the imputation method, and applied the spectral clustering method for the single-cell clustering analysis. From the reviewer's point of view, this method development seems a bit weak and lacks novelty for a full computational paper. In addition, the SCASL software has issues with using it, described in the 'code and software review' section. SCASL cannot be used for broader single-cell data generated from droplet-based platforms, e.g., 10x genomics. This would need a major improvement of the computational analysis pipeline.

From the biological perspective, although reported in previous studies (Song et al., Mol Cell 2017; Olivieri et al., eLife 2021; Wen et al., NAR 2023; Benegas et al., eLife 2022; Liu et al., Sci. Rep. 2021; Buen Abad Najar et al., Genome Res. 2022; Wang et al., Sci Adv. 2022), the manuscript showed interesting cases in diverse biological contexts wherein AS demonstrates the advantages of single-cell analysis. Nevertheless, while the manuscript introduces interesting examples, the claims such as "precancerous/early tumor cell stages" and "novel class of cells with potential crosstalk" have no support or validation from more detailed analysis and the experimental results, leading the paper to a more hypothesis-driven and descriptive explanation rather than a rigorously validated one.

More concerns are described below.

Major points

- It would enhance the introduction section if the authors added more relevant information from the existing tools/methods for AS analysis, such as BRIE, Expedition, MARVEL, SpliZ, Psix, etc. The introduction lacks many relevant references in this research field.
- The manuscript lacks comparison/benchmarking against the existing tools/methods developed in this research field. In particular, SCASL relies on LeafCutter for the AS quantification; how this quantification approach differs from other existing tools? how do different types of AS quantification approaches affect the downstream cell clustering results?
- The authors demonstrated the robustness of the imputation method but did not show the clustering analysis results compared with and without imputation. It seems the dropouts do not much affect the clustering analysis when the authors randomly removed the large proportions of the AS sites (Figs. S1C and D). If that is the case, is the imputation still required for a dataset that

has low dropouts?

- How the performance of the proposed imputation technique compares to other existing imputation approaches?

- It is not clear for the comparison of AS vs. expression UMAP and the correlation heatmap (e.g., Figs. 2A, 2B, 2E, 2F). How many AS and genes (variable features), and PCA dimensions were used? Were these UMAP and heatmap generated in a fair comparison? For example, were they from the same genes?

- For the first breast cancer dataset, from the AS analysis, the C2 cluster was identified as an early intermediate stage during the malignant transformation. How would the authors verify this? This would need additional information, e.g., the presence of mutations/copy number changes in these cells that are required to confirm this. In addition, more trajectory analysis results, e.g., monocle, PAGA, and RNA velocity-based analyses, would be helpful to visualize cellular trajectories objectively.

- For the second breast cancer dataset, Fig. 3B, why is there no UMAP from gene expression to make a comparison with AS, similar to Figs. 2A and B? It seems using only gene expression, in this case, would be enough to capture the C3 cluster. Similar to the above comment, how would the authors verify that the C3 cluster is the precancerous cells? The descriptive information from differential gene expression would not be enough. Would these precancerous have some additional information, e.g., the presence of mutations/copy number changes? More trajectory analysis results from different methods would be required. This would need supporting experimental validation to prove the link between the shift of splicing landscapes and the precancerous cells.

- There is no comparison between AS and gene expression for the dataset shown in Fig. 4a. Would the two cellular trajectories be reproducible using the gene expression? More objective visualization from multiple trajectory analyses would be required. Where were the arrows derived from in Fig. 4a? There is no description of what they mean in the legend.

- There is a claim of reporting "a novel class of cells that potentially function in the crosstalk during embryonic liver development". What it means by "crosstalk" in this context? Probably this needs to tone down since no experimental results support this. Would this C4 cluster be identified using gene expression?

- It is not clear for the comparison of AS vs. expression UMAP in Fig. S7. How many AS and genes (variable features) and PCA dimensions were used? Were these UMAP generated in a fair comparison? For example, were they from the same genes? Also, it is difficult to visually inspection of the comparison by looking at many labeled colors in UMAPs. This would need some other better plot types to summarize the comparison results. Would the Treg clusters (C3 and C4) be differentiated if used gene expression of the same genes as in AS?

- The manuscript describes a lot of descriptive information on T-cells analysis. This section is not concise, and some explanations are probably not informative and relevant to the manuscript, such as Fig. S10B.

- The concept that each single cell has just one dominant splicing form is not quite right. Recent studies using long-read sequencing revealed genes with multiple isoforms in a single cell (Volden et al., 2022; Tian et al., 2021; Mincarelli et al., 2023). The detection of isoforms per gene per single cell varies across genes and expression levels.

- Is the AS probability matrix tolerant to the batch/donor effect? If not, how do the authors deal with this for the downstream analyses, e.g., clustering and differential splicing analysis?

- In the discussion, it should be clearly stated that the SCASL cannot be used for the droplet-based single-cell data.

Minor points

- Gene names in texts and figures must be in italics.
- Table 1 shows only 5 datasets, not 6. A dataset from Ref. 93 seems to be missing from the table and main text.
- TRDJ1 is missing in Fig. 5B.
- Why is the representation inconsistent across the manuscript, e.g., UMAP in Fig. 2 and TSNE in Figs. 3 and 4?
- Why two TSNE plots in S4 are different?
- The correlation scale should be consistent across graphs. (e.g., Figs 2E, F and 3B, C, and others).
- No statistics on how many AS events were identified and how many genes they belonged to. It would be useful to see the distribution of detected AS event types (e.g., exon skipping, 5' and 3' alternative splicing, etc.). How do distinct AS event types contribute to the clustering analysis?
- The figures showing differential variance results (e.g., Figs. 5C, 6A, B, D) lack statistical information (e.g., p-values).

Code and software review

- There is a strong lack of documentation for the tool on how to produce the input, run the tool, and control the parameters. For example, the main documentation states that the tool uses BAM files as the initial input, but from running and inspecting the code, it's clear that it also requires computed junction sites derived from BAM files. Also, no details are provided on how to derive these junctions, and the default example run is not working without additional edits to the config file.
- The interpretation of the results also lacks documentation, e.g., the meaning of the plots, result file formats and values, etc. In case a person is interested in using intermediate files, it will be hard to figure out what each intermediate output means.
- In the file "config/srr_demo.yaml", it should be "junction : data/junction" not "junction : data/junc" and "label_file : data/label.csv" should be "label_file: data/Demo_label.csv".
- The installation of the software has a problem. The reviewer successfully installed the SCASL pipeline only on the Linux system using the Conda environment. However, it never success on macOS. This is likely because the required LeafCutter software cannot be installed in R (tested on different versions of R) on macOS. See the error below. The reviewer did not try it on the Windows system. Since SCASL relies on the LeafCutter, this would also be a problem for the SCASL.

```
/Library/Frameworks/R.framework/Versions/4.0/Resources/library/StanHeaders/include/stan/math/prim/core/init_threadpool_tbb.hpp:12:10: fatal error: 'tbb/task_scheduler_init.h' file not found
#include <tbb/task_scheduler_init.h>
^~~~~~
28 warnings and 1 error generated.
make: *** [stanExports_bb_glm.o] Error 1
ERROR: compilation failed for package 'leafcutter'
* removing '/Library/Frameworks/R.framework/Versions/4.0/Resources/library/leafcutter'
Warning message:
In i.p(...) :
installation of package
'/var/folders/c6/_w309xr54n7cphf9nw4x_ms40000gn/T//RtmpRm3CpN/file1cb049fa7b6b/leafcutte
r_0.2.9.tar.gz' had non-zero exit status
```

In file included from
/Library/Frameworks/R.framework/Versions/4.0/Resources/library/StanHeaders/include/stan/math/prim/core.hpp:4:

executing sites quality filter by threshold

the site histogram is saved at X_process_result/20230807125224/img/site_hist.png

the descriptions of the non-NaN data of sites are shown below

count 5885.000000

mean 6.037553

std 4.070685

min 4.000000

0% 4.000000

10% 4.000000

20% 4.000000

30% 4.000000

40% 4.000000

50% 4.000000

60% 5.000000

70% 6.000000

80% 7.000000

90% 10.000000

max 36.000000

dtype: float64

the site histogram is saved at X_process_result/20230807125224/img/site_hist.png

the descriptions of the non-NaN data of sites are shown below

count 5803.000000

mean 5.954679

std 4.006180

min 4.000000

0% 4.000000

10% 4.000000

20% 4.000000

30% 4.000000

40% 4.000000

50% 4.000000

60% 5.000000

70% 5.000000

80% 7.000000

90% 10.000000

max 37.000000

dtype: float64

done.

remove the duplicated site starts and ends...

done.

executing sample quality filter...

the sample histogram is saved at X_process_result/20230807125224/img/sample_hist.png

the descriptions of the non-NaN data of sites are shown below

count 37.000000

mean 1893.756757

std 2963.414457

min 482.000000

0% 482.000000

10% 651.200000

20% 748.800000

30% 922.400000

40% 974.400000

50% 1068.000000

60% 1105.800000

70% 1225.400000

80% 1408.600000

90% 1663.200000

max 11684.000000

dtype: float64

```
done.
saving...
done.
=====Normalization & Imputation=====
reading data from X_process_result/20230807125224/filtered_matrix...
Traceback (most recent call last):
File "main.py", line 10, in <module>
scasl.fit()
File "SCASL-main/scasl/splice.py", line 95, in fit
run_cluster(self.cfg)
File "SCASL-main/scasl/splice.py", line 66, in run_cluster
df_final, mat = normalize(filter_path, cfg.impute.num_iteration, cfg.impute.knn)
File "SCASL-main/scasl/normalize.py", line 92, in normalize
dfs = norm_only(df_path, 'start')
File "SCASL-main/scasl/normalize.py", line 40, in norm_only
df_prob = to_prob(df, groupby=groupby)
File "SCASL-main/scasl/normalize.py", line 22, in to_prob
sums = sums.drop(columns=['start', 'end'])
File "/.conda/envs/scasl/lib/python3.9/site-packages/pandas/core/frame.py", line 5258, in drop
return super().drop(
File "/.conda/envs/scasl/lib/python3.9/site-packages/pandas/core/generic.py", line 4549, in drop
obj = obj._drop_axis(labels, axis, level=level, errors=errors)
File "/.conda/envs/scasl/lib/python3.9/site-packages/pandas/core/generic.py", line 4591, in
_drop_axis
new_axis = axis.drop(labels, errors=errors)
File "/.conda/envs/scasl/lib/python3.9/site-packages/pandas/core/indexes/base.py", line 6699, in drop
raise KeyError(f"{list(labels[mask])} not found in axis")
KeyError: "['end'] not found in axis"
```

Reviewer #3 (Remarks to the Author):

I co-reviewed this manuscript with one of the reviewers who provided the listed reports as part of the Nature Communications initiative to facilitate training in peer review and appropriate recognition for co-reviewers

Reviewer #4 (Remarks to the Author):

I co-reviewed this manuscript with one of the reviewers who provided the listed reports as part of the Nature Communications initiative to facilitate training in peer review and appropriate recognition for co-reviewers

Reviewer #5 (Remarks to the Author):

Xiang and colleagues presented a novel single-cell alternative splicing analysis method, that capable of conducting single-cell clustering based on alternative splicing and identifying previously undiscovered subgroups that are not detectable through single-cell gene expression clustering. Initially, the probabilities of the 5' and 3' ends were calculated as indicators of the degree of alternative splicing (AS). Subsequently, an iterative kNN (k-Nearest Neighbors) approach was employed to obtain AS probabilities for events with low sequencing depth and dropout events. These imputed AS probabilities were marked to reflect the influence of gene expression to a certain extent. Finally, a cluster analysis based on spectral clustering was conducted on the resulting matrix to delineate the alternative splicing landscape of single cells. Applying the SCASL approach to five datasets pertaining to disease and development, a series of novel subpopulations associated with alternative splicing were identified.

However, I have some concerns about certain aspects of the manuscript. In scRNA-seq clustering based on alternative splicing, three main challenges exist: low sequencing depth, dropout events, and interference from gene expression levels (with or without expression in the cell). While SCASL addresses the impact of sequencing depth and dropout events through imputation, it does not explicitly explain how to mitigate the influence of gene expression.

1. The author employs the iterative kNN method to impute AS probabilities; however, there are concerns regarding the calculation indicator used for kNN. It appears that the Euclidean distance is calculated based on the probabilities of all AS events in cells, but it is crucial to consider whether the calculation indicator should be based on the probabilities of all AS events of the gene in the cell, rather than all genes.

2. The clustering is directly performed based on the imputation probabilities and indicator vectors, which may not be optimal due to NA values in the AS probabilities matrix for genes not expressed in the cell. To address this, the author should construct the real AS probabilities matrix using both NA values and non-NA values (AS probabilities). Additionally, clustering based on cell-gene NA values and non-NA values could resolve the heterogeneity present in cells, as demonstrated in the study "Embracing the dropouts in single-cell RNA-seq analysis, 2019". Therefore, the author needs to address three key problems: firstly, how to construct the real AS probabilities matrix; secondly, how to calculate the proximity distance between cells based on the AS probabilities matrix under such complex situations; and lastly, how to distinguish the influence of gene expression from alternative splicing during clustering based on the AS probabilities matrix in complex scenarios.

3. Regarding batch effects, it would be valuable to understand how the author perceives their impact and how SCASL accounts for them during clustering based on alternative splicing.

4. There are a few specific questions that require clarification. For instance, further explanation is needed on the calculation of the AS probabilities matrix and the construction of a $4N * M$ matrix for clustering. Additionally, the roles of upstream-related grouping, downstream-related grouping, and indicator values in the $4N * M$ matrix during clustering need elaboration.

5. Previous studies have demonstrated that the alternative splicing state of T cells has limited efficacy in distinguishing between CD8 and CD4 T cells. The question arises whether SCASL incorporates additional factors, such as gene expression, that contribute to their complete separation (Figure 5A and Figure S8B).

In conclusion, while SCASL offers promising solutions for scRNA-seq clustering based on alternative splicing, the manuscript should address the mentioned concerns and provide further clarity on certain aspects of the methodology.

Responses to the Reviewers' Comments

We would like to thank the reviewers for their time and efforts in reviewing our manuscript. We are glad to see that all the reviewers showed great interests in our study and provided highly positive feedbacks. The reviewers have made very thoughtful suggestions and comments, which greatly helped us further improve the manuscript. We now have added a series of analyses and more discussions to address the questions. Please refer to the following point-by-point responses for the details.

Reviewer #1 (Remarks to the Author):

The authors develop a pipeline "SCASL" to cluster cells based on alternative splicing (AS), rather than total gene expression. Their pipeline is roughly LeafCutter -> KNN smoothing -> spectral clustering. They show various examples in Smart-seq data where it seems SCASL is pulling out biologically meaningful clusters. The paper is reasonably clearly written and the methods are reasonable overall. I have two main concerns with the paper however.

Main concerns:

First is that for a paper presenting a novel method there is a lack of systematic evaluation. The biological findings are nice but don't really address whether 1) SCASL is really finding more robust or meaningful clusters than gene expression and 2) whether the various choices being made in the pipeline are optimal (e.g. how is sigma chosen in KNN? Does the iterative part of KNN actually improve performance). For (1), it is of course difficult to assess clustering quality. But I think something like (repeatedly) randomly splitting a dataset in two (two sets of cells) and then assessing the reproducibility of the clusters found using SCASL vs Seurat (applied to each subdataset separately) would be a good start.

Response: We thank the reviewer for the thoughtful questions and suggestions above. We have added a series of new analyses for a more systematic evaluation of the pipeline.

1) Robustness of SCASL. We performed random sub-sampling of the cells (100%, 75%, or 50%), which were then used for cell clustering based on their splicing or gene expression profiles (Fig. S2). For the group of 100% sub-sampling, clustering procedures were simply repeated with the full datasets, and the results provide an assessment of robustness when different random seeds were used by the computer. The clusters of the sampled cells were then compared to the original results reported in the previous version of the manuscript. As shown in Fig. S2C (reproduced as follows), for different ratios of sub-sampling, the cell clusters defined by SCASL based on single-cell AS profiles are much more robust than the ones defined by Seurat based on the gene expression profiles. When repeated with the full datasets, SCASL also demonstrated greater stability compared to the expression profile-based method.

The box plot shows the comparison of each clustering result with the original clustering result when 100%, 75%, and 50% of cells were randomly sampled multiple times in the hepatoblast data set and TNBC-2 data set. The AS profiles were used by SCASL and the gene expression profiles were used by Seurat for the clustering analyses.

2) Selections of the parameters. We are sorry for not making this clear. A subsection of parameter selection has been added into the revised Methods section (page 28-29). In brief, for the optimal number of cell clusters, we have provided the elbow method in SCASL. Users also have the option of assigning any specific number of clusters.

For the KNN process, first, we used the average distance to the k nearest neighbors as the value of sigma for imputation, which is a common practice in most, if not all, tools based on KNN imputation, e.g., missForest [1], VIM [2], etc. Second, determinations of other iteration parameters, i.e., the number of neighbors k and the number of iterations t, requires comprehensive consideration of sample size, data sparsity, and computation costs. Based on our analysis with multiple datasets, the imputation tends to converge after three iterations (shown in the following figure).

The line chart illustrates the mean absolute difference between the true value and the imputed value during different iterations of the imputation process. 4 datasets, listed to the right, were used.

The selection of the parameter k primarily depends on the size of the dataset. According to the rule of thumb, k is typically set as the square root of the total number of samples or a smaller fixed value[3]. Large k values result in imputations based on too many neighbor cells, which would lead to underestimations of cellular heterogeneity, whereas small k values tend to introduce stochastic noise due to lack of enough reference for imputation. Additionally, the choice of k has a significant impact on the computation cost for large datasets. A smaller k value can greatly improve the imputation speed. After systematic comparisons, we have set the default value of k to 10, which is suitable for a wide range of data with sample numbers ranging from hundreds to thousands.

In summary, according to our tests with more than 10 different published datasets, the default parameters ($k=10$ and $t=3$) work quite well in data imputation and cell clustering. However, in special cases such as extremely large datasets or abnormally high sparsity, users can easily adjust the parameters to compare the outcomes and get the optimal results, just like what we are doing with most bioinformatics algorithms.

Second is the addition of the missingness matrix. This sounds perfectly reasonable, but it is very possible that it acts as a proxy for expression. Junctions that are missing in a sample are indicative of low expression of the gene, and vice versa. Thus this missingness matrix is effectively binarized total expression. The smoothing and clustering are therefore getting information about BOTH AS and expression. That's not necessarily a bad thing: indeed to get the best possible cell type resolution this might be very sensible. This needs exploring: can SCASL still cluster at all without the missingness matrix? If not that's OK, but the description of the method should acknowledge this potential issue.

Response: We thank the reviewer for the insightful comment. The missingness matrix indicates the number and positions of the missing values of splicing junction probabilities, which were shown as NA in the input data. Therefore, the missingness matrix (also known as the NA indicator matrix) indeed reflects the overall RNA abundance to some extent, albeit still being different from the gene expression profile.

As suggested by the reviewer, we tested the performance of SCASL without the information of missingness, by either completely removing the missingness matrix or shuffling the binary missingness values independently for each cell. As the reviewer has suspected, both operations resulted in significantly different results of cell clustering (Fig. S14, also reproduced as follows). For most of the datasets, removing or shuffling the missingness matrix would largely disrupt the cell clusters that we obtained before with SCASL. These new clusters fail to recapitulate important cellular heterogeneities between tumor vs normal cells, primary vs metastatic cells, and different cell development lineages. Therefore, the missingness matrix is indispensable for definitions of cell clusters by SCASL based on AS.

In addition, it is worth noting that the NA shuffling process above does not alter the number of the missing values in each cell. Therefore, the clustering results demonstrate that the positions of NAs, rather than their numbers (i.e., overall sparseness), contributes to the clustering of cells.

On the other hand, the missingness matrix alone is certainly not informative enough to define the cell clusters. The percentages of missing AS probability values (values of NA) are comparable across the cell clusters defined by SCASL (Fig. S15, also reproduced as follows). Furthermore, upon shuffling of the AS probability matrix, SCASL also failed to recover the cell clusters with biological relevance (Fig.

S14, reproduced as follows). Therefore, a critical and necessary feature of SCASL is to incorporate both the AS profile and the data missingness information for defining cell clusters. In other words, both the information of AS probabilities and data sparsity pattern contributes to the cell heterogeneity landscapes as portrayed by SCASL. The results above have been added into the revised manuscript.

Clustering with only AS information

Clustering results using only the AS information. For each of the 6 datasets, the figure in the top row shows the clustering results from the complete process of SCASL (AS and NA information), and the figures in the bottom row shows the results of using only the AS information.

Figures on the left show the clustering results using the complete process of SCASL (AS and NA information). Figures in the middle show the clusters defined by SCASL with the AS profile shuffled. Figures on the right show the clusters defined by SCASL with the missing data information shuffled.

The percentages of missing AS probability values (NA values) across the cell clusters defined by SCASL are shown in box plots.

More minor issues:

SCASL uses junction usage relative to other junctions that share either the same 3' or 5' splice site (so each junction will actually be represented twice). This is different from the LeafCutter approach of forming junction "clusters". FRASER also introduced a third option which is to compare each junction to the union of all junctions with shared splice sites. It's not clear to me which approach is better, and it's possible different approaches would be better in different contexts, e.g. I believe the FRASER authors argue their approach is particularly appropriate for missplicing caused by rare variants/mutations.

Response: We thank the reviewer for pointing this out. In fact, the method we used to quantify AS events was quite similar to FRASER. Specifically, AS events sharing the same 5' SSs or 3' SSs were grouped together, and then for each AS event, the junction read count was divided by the count of all the junction reads from all the AS events within the group. Here, we used the R package of LeafCutter just to extract and count the junction reads. We did not follow the method of LeafCutter for downstream quantification of the AS events. We are sorry for the misleading statement of using LeafCutter package in the previous version. We have modified the method section to more clearly describe the quantification of AS probabilities (page 26-27).

As the article of FRASER has claimed [4], which we totally agree, the strategy we used for quantifications of AS events does not rely on prior annotations of exons. Its results provide a comprehensive survey of different splicing patterns, including the canonical AS events (such as exon skipping, intron retention, mutually exclusive exons) as well as other splicing errors due to rare mutations or aberrant spliceosome.

Using AS to discover additional heterogeneity beyond that represented by expression alone has explored before (e.g. <https://elifesciences.org/articles/70692> and <https://academic.oup.com/nar/article/51/5/e29/6985826>, which are not cited/compared to).

Response: We thank the reviewer for pointing out these previous studies in the related field, which indeed worth discussion (added on page 3-4). However, the major goals and methodology designs of these two studies are different from SCASL. MARVEL[5] was designed to identify differential splicing events between different groups of cells. It integrates gene expression and alternative splicing profiles into a single analysis framework to evaluate the correlation between differential splicing events and gene expression changes. Cell types have to be predefined as an input of MARVEL.

SpliZ[6] provides a new way of quantifying the AS levels of genes. It was based on PSI and used Z-score values to measure how deviant a cell's splicing is compared to a population average. It was not designed for cell clustering based on splicing landscapes. However, we took the liberty of using the SpliZ scores for systematic clustering of the cells. As shown in the following figures, the cell clusters do not show biological relevance based on prior knowledge of the cell types.

Nevertheless, compared to the existing tools, SCASL was designed for a very different purpose of single-cell splicing analysis, i.e., clustering of single-cells and recapitulating the cell identified defined by their global splicing landscapes. Other methods such as MARVEL and SpliZ would be helpful for

further downstream analyses of the biological relevance of these cell subtypes in various physiological contexts.

Cell clustering results of SCASL and SpliZ. Cell clustering results of the same SCASL pipeline but with AS probabilities quantified by junction reads (left) or SpliZ scores (right).

It would be interesting to know if SCASL can do anything with 10x data despite it's substantial 3' bias.

Response: We thank the reviewer for the intriguing question. We tested the performance of SCASL on a dataset generated by 10x platform for dentate gyrus cells [7] (in mice at postnatal days 12, 16, 24, and 35). It is known that neuroblasts differentiate into granule cells. Clustering of neuroblasts and granule cells can be challenging, as both cell types undergo developmental transitions and share certain molecular markers. The cell clusters defined by SCASL based on 10x RNA-seq data indeed

show patterns indicating potential differentiation transitions (Fig. S16A, B), which are supported by differentiation marker genes. By contrast, such transitions were not recapitulated by the cell clusters defined by their gene expression profiles, which only shows two major groups of neuroblasts and granule cells. It should be noted that the cell type annotations were defined based on gene expression profiles in the original study. Therefore, it is well expected to see the two types of cells clearly separated in Fig. S16C.

This analysis shows that despite the strong 3' bias, 10X RNA-seq data can also be used for cell clustering by SCASL. As discussed in the manuscript, the information of cell identities supplied by AS profiles are highly redundant, and therefore, fractions of the AS profiles within the 3' regions of mRNAs could be informative for definitions of cell clusters. Technically, the procedure of SCASL is exactly the same with either SMART-seq or 10X data. However, it should be noted that most of the 10X RNA-seq libraries were subjected to very low depth of sequencing. Therefore, splicing-based clustering with SCASL is not recommended for the droplet-based 3' end RNA-seq data with low sequencing depth.

Results of SCASL applied on 10X data of Dentate gyrus cells. (A) TSNE plot showing clustering of 2442 cells by SCASL based on the AS landscapes. The cells are color labeled by the clusters defined by SCASL (left) and cell types defined by marker genes (right). The principal component analysis (PCA) is performed with a dimensionality reduction number of 20. (B) The dot plot shows

the expression of marker and functional genes in poor differentiation (Neuroblast) and well differentiation (Granule cell). The dot sizes represent the proportions of cells with expression, and the color scale represents the mean expression level. (C) TSNE plot showing clustering of 2442 cells by Seurat based on the gene expression profiles with 3000 variable features. The cells are color labeled by the clusters defined by SCASL (left) and cell types defined by marker genes (right). The principal component analysis (PCA) is performed with a dimensionality reduction number of 20.

I'm a bit concerned about the FET for differential splicing. RNA-seq read counts (including those from UMI-less Smart-seq) are typically overdispersed (in other words each read is not independent), which FET cannot account for. Something like a beta-binomial GLM would be a better alternative.

Response: We believe that this is a misunderstanding, and we are sorry for not making this clear. In fact, due to the overdispersed reads, we did not use RNA-seq read counts directly for differential splicing analysis between different cell groups. Instead, considering the highly polarized AS probabilities in single cells, we turned the differential splicing analysis into a classification problem. Specifically, for each AS event, we asked whether its occurrence is preferentially enriched in one cell cluster but not in another cluster. Here, an AS event with probability higher than 0.6 is categorized as present in the cell, whereas an event with probability lower than 0.4 is categorized as absent. In fact, the probabilities of most AS events classified as absent were close to 0, whereas the ones classified as present have probabilities close to 1. After such categorizations, we then performed the Fisher's Exact Test for pair-wise comparisons between cell clusters to evaluate whether an AS event is preferentially enriched in one cell cluster but depleted in the other one. The text above has been added into the revised manuscript (page 31-32).

Spectrum -> spectral clustering

Response: We are sorry for the typo and thanks for pointing it out. The manuscript has been proof-read for several times to correct the typos.

Fig 2c: do you see this relationship if you cluster by expression?

Response: We do not see the relationship in the clusters defined by gene expression profiles (Fig. S4B). The results have been added into the revised manuscript.

Pseudo-time analysis of the gene expression clustering results was performed using CytoTRACE. The clustering labels on the left side are derived from the original paper of the dataset, while the clustering labels on the right side correspond to the clustering results obtained from Seurat based on gene expression (with the same number of clusters as the AS clusters).

For the TNBC I imagine you could detect somatic mutations from the Smart-seq data: it would be interested if they correlated with your clusters. Maybe beyond of the scope of the current paper however.

Response: We thank the reviewer for the thoughtful suggestion. We have added analyses of somatic mutations for the TNBC cell clusters. We used Monovar [8] for detections of single nucleotide variation (SNV) from the SMART-seq reads of single cells. For the first TNBC study, the cells in C1 bear significantly more SNVs than the cells in C2 (Fig. S4E). This is nicely consistent to our hypothesis that the cluster of C2 represents an early stage of tumor cells, whereas C1 represents a more malignant stage. By contrast, the cluster of C0, which mainly consists of the cells from micro-metastatic sites, exhibited slightly elevated rates of mutations, indicating that micro-metastasis takes places during early stage of tumor development. This is consistent to the previous notes [9, 10].

Finally, it should be noted that due to lack of non-cancerous cells from the same patient, these SNVs were identified by alignment of the RNA-seq reads with the human genome reference. They are not all somatic mutations. Nevertheless, the general trend of SNV numbers in the three clusters supports the proposed cellular patterns of tumor development revealed by SCASL based on the AS landscapes.

Numbers of SNVs in the single-cells inferred using MonoVar.

Your representation is not "4 dimensional". It's 2D. It's also weird to consider the matrix as 4NxM rather than Nx4M.

Response: This is indeed a misstatement. The representation is certainly 2D. We have rewritten this part of the method (page 29).

Reviewer #2 (Remarks to the Author):

In this manuscript entitled “Interrogations of single-cell RNA splicing landscapes with SCASL define new cell identities with physiological relevance”, the authors developed SCASL, a new approach for single-cell clustering based on alternative splicing (AS) information. The pipeline relies on the LeafCutter software to identify AS events. SCASL identifies the AS sharing 5' and 3' splice sites and quantifies AS probability values. Due to high dropouts in single-cell data, the author developed the imputation approach for missing values and applied the spectral clustering approach for clustering analysis.

Mainly described in the manuscript, the authors applied SCASL to analyze publicly available single-cell datasets generated from the SMART-Seq protocol and demonstrate the method's capability. The manuscript shows, in some of the datasets, a better single-cell clustering for identifying cell identities using AS than standard gene expression.

The proposed method generally represents an important step forward using the AS information as another layer for identifying cell identity. However, there are significant concerns (see below). In particular, it is not clear to the reviewer whom this manuscript is written for, a biologist or a bioinformatician/computational biologist.

Response: We thank the reviewer for the positive comments for our work. The vast majority of the previous analyses of transcriptome heterogeneity were simply based on single-cell RNA expression data. The present study is positioned as development of a new method for interrogating the splicing heterogeneity. The underlying assumption of our method is that the RNA splicing landscapes bear extensive heterogeneity and represent a different layer and aspect of cell identity, which is different than the cell clusters defined based on gene expression profiles. Therefore, we applied our newly developed method on different datasets, to better promote the notion that the AS landscape reflects

a different type of cell heterogeneity with biological and physiological relevance.

From the computational perspective, the manuscript lacks details on the new algorithms and benchmarking, particularly for comparing with existing methods/tools. The authors developed the AS probability matrix built on LeafCutter's results, implemented the imputation method, and applied the spectral clustering method for the single-cell clustering analysis. From the reviewer's point of view, this method development seems a bit weak and lacks novelty for a full computational paper. In addition, the SCASL software has issues with using it, described in the 'code and software review' section. SCASL cannot be used for broader single-cell data generated from droplet-based platforms, e.g., 10x genomics. This would need a major improvement of the computational analysis pipeline.

Response: The reviewer has raised 4 comments about 1) details on the algorithm and benchmarking by comparing with other methods, 2) dependence on other tools and novelty, 3) technical issues for running the pipeline, and 4) applications on droplet-based data. Please refer to our detailed point-by-point responses to the "Major points" as follows.

From the biological perspective, although reported in previous studies (Song et al., Mol Cell 2017; Olivieri et al., eLife 2021; Wen et al., NAR 2023; Benegas et al., eLife 2022; Liu et al., Sci. Rep. 2021; Buen Abad Najar et al., Genome Res. 2022; Wang et al., Sci Adv. 2022), the manuscript showed interesting cases in diverse biological contexts wherein AS demonstrates the advantages of single-cell analysis. Nevertheless, while the manuscript introduces interesting examples, the claims such as "precancerous/early tumor cell stages" and "novel class of cells with potential crosstalk" have no support or validation from more detailed analysis and the experimental results, leading the paper to a more hypothesis-driven and descriptive explanation rather than a rigorously validated one.

Response: We have provided further analyses to better support the biological insights provided by the new scheme of AS-based cell clustering. The text of the manuscript has also been modified to describe the biological insights more precisely. Please refer to our point-by-point responses below.

Major points

1. It would enhance the introduction section if the authors added more relevant information from the existing tools/methods for AS analysis, such as BRIE, Expedition, MARVEL, SpliZ, Psix, etc. The introduction lacks many relevant references in this research field.

Response: We thank the reviewer for the suggestion. Indeed, there have been a series of analytical methods for mining of biological information from different aspects of RNA splicing profiles. However, none of them was designed specifically for a systematic interrogation of the AS heterogeneity and clustering of single-cells. We have added a paragraph into the Introduction section to introduce these existing methods as a background of our work (page 3-4).

The existing approaches for quantifying alternative splicing (AS) levels were based on either PSI (percent splice-in) or junction reads. Most of these AS analysis tools primarily focus on identifying differential splicing events. For instance, BRIE[11] (for bulk data) and Expedition[12] (for single-cell

data) employ Bayesian models to estimate PSI for differential splicing analysis, while rMATS[13] (for bulk data) employs a linear mixed-effects model. Psix[14] quantifies AS using PSI values and identifies splicing events associated with cell state through autocorrelation models. On the other hand, Leafcutter[15] identifies junction reads for definitions of AS events and then adopts a generalized linear model to quantify the AS events. FRASER[4] uses a different strategy for identifying and quantifying the AS events from junction reads. MARVEL[5] relies on PSI defined by rMATS and junctions reads to define single-cell AS and then integrates gene expression data to analyze the effects of differential splicing between cell groups. Additionally, newer methods have emerged that define AS levels at the gene level rather than for specific AS events, such as SpliZ[6]. Nonetheless, these tools typically require known cell labels for differential analysis, and methods specifically designed for unsupervised clustering based on single-cell splicing landscapes are urgently needed.

2. The manuscript lacks comparison/benchmarking against the existing tools/methods developed in this research field. In particular, SCASL relies on LeafCutter for the AS quantification; how this quantification approach differs from other existing tools? how do different types of AS quantification approaches affect the downstream cell clustering results?

Response:

1) It is a misunderstanding that we simply used LeafCutter for AS quantification, and we are sorry for not making this clear. Our approach aligns with the concept put forth by Leafcutter, which utilizes junction reads to represent AS. However, the method for quantifying the AS events was different than LeafCutter. Specifically, AS events sharing the same 5' SSs or 3' SSs were grouped together, and then for each AS event, the junction read count was divided by the count of all the junction reads from all the AS events within the group. Here, we used the R package of LeafCutter just to extract and count the junction reads. We did not follow the method of LeafCutter for downstream quantification of the AS events. The subsequent steps involving grouping, filtering, and ratio calculation of AS events were different and coded in SCASL independently. We are sorry for the misleading statement of using LeafCutter package in the previous version. The method section has been modified (page 26-27).

2) The strategy we used for quantifications of AS events does not rely on prior annotations of exons. Its results provide a comprehensive survey of different splicing patterns, including the canonical AS events (such as exon skipping, intron retention, mutually exclusive exons) as well as other splicing errors due to rare mutations or aberrant spliceosome.

Another strategy for AS quantification relies on prior annotations of exons. Under this category, PSI (percent spliced-in), has been widely used to quantify the ratio of transcripts skipping a specific exon. PSI performs exon-by-exon analysis, and it is inefficient in identifying intron retention and other complex AS events [5]. Furthermore, SpliZ is another popular tool for AS analysis. Notably, SpliZ assesses splicing differences at the gene level rather than focusing on specific AS events.

First, the computational costs for AS quantification are quite different across these three methods. The method based on junction reads is the most efficient one, whereas PSI takes the longest time.

Comparison of the time consumption. The box plot displays the time consumption of quantitative AS analysis using junction reads (SCASL), SpliZ score (SpliZ), and PSI (Outrigger). 3 datasets were used for the tests.

As suggested by the reviewer, we have compared the performances of SCASL based on these three ways of AS quantification. As shown in the results of three datasets, neither PSI nor SpliZ could result in efficient definitions of cell clusters with biological relevance based on prior knowledge of the cell types. Therefore, we believe that the method design of SCASL is best suited for the current strategy of AS quantification, i.e., percentages of junction reads.

Clustering by different AS quantification methods

Cell clustering results of SCASL based on different quantification methods of AS. Cell clustering results of the same SCASL pipeline but with AS probabilities quantified by junction reads (left), SpliZ scores (middle), or PSI (right).

3. The authors demonstrated the robustness of the imputation method but did not show the clustering analysis results compared with and without imputation. It seems the dropouts do not much affect the clustering analysis when the authors randomly removed the large proportions of the AS sites (Figs. S1C and D). If that is the case, is the imputation still required for a dataset that has low dropouts?

Response: The imputation is needed to fill the spots of missing AS probability values. Single-cell RNA-seq data usually have low sequencing depth and coverages. To ensure that the AS probability

estimation is accurate and reliable, we filtered out a significant number of splicing sites with low read depths (few than 10 junction reads for a group of AS events sharing the same 5' or 3' splicing site). Therefore, estimations of the AS probabilities for large numbers of splicing sites are missing in the AS probability matrices, which are marked as NA. These NA values cannot be simply converted to 0, which actually means skipping of the particular splicing event. The downstream spectral clustering cannot handle such matrices with missing values, which necessitates the procedure of data imputation.

4. How the performance of the proposed imputation technique compares to other existing imputation approaches?

Response: We appreciate the reviewer's suggestion. In fact, we selected KNN after a comparison among several different strategies for imputation as follows:

- 1) Mean / median imputation.
- 2) K-nearest neighbors (KNN) imputation.
- 3) Model-based imputation: Markov affinity model, dropouts model, etc.

Specifically, in addition to KNN, we have also tested the method of mean value imputation and two widely utilized model-dependent imputation methods, MAGIC [16] (Markov Affinity-based Graph Imputation) and scImpute [17] (imputation based on dropouts model). For all the methods, the same forced data dropout was performed so that they all processed the same testing datasets, and the deviations between the predicted and true values are summarized in the following figure. It is quite clear that KNN outperforms all the other methods.

Deviation between imputation and true value

Comparison of different imputation methods for AS profiles. (A-D) Violin plots showing the difference between true and imputed values at different ratios of forced data dropouts. **(A, B)** Results of 3 datasets (brain, macrophages, and hepatoblasts) processed by SCASL (A) and MAGIC (B). **(C)** Results of one datasets as an example processed by imputation based on mean values. **(D)** Results of two datasets processed by scImpute.

5. It is not clear for the comparison of AS vs. expression UMAP and the correlation heatmap (e.g., Figs. 2A, 2B, 2E, 2F). How many AS and genes (variable features), and PCA dimensions were used? Were these UMAP and heatmap generated in a fair comparison? For example, were they from the same genes?

Response: We have added the information of gene numbers used for the UMAP and correlation

heatmaps into the Figure Notes and Methods section. In general, both the analyses based on AS and expression are transcriptome-wide and unbiased, and the lowly expressed genes were removed for both AS and expression analyses. Eventually, although the total number of genes used are slightly different in the AS and expression analyses, they are roughly the same group of genes with overlaps usually higher than 90%. Therefore, the comparisons between AS vs expression features are fair and reasonable.

6. For the first breast cancer dataset, from the AS analysis, the C2 cluster was identified as an early intermediate stage during the malignant transformation. How would the authors verify this? This would need additional information, e.g., the presence of mutations/copy number changes in these cells that are required to confirm this. In addition, more trajectory analysis results, e.g., monocle, PAGA, and RNA velocity-based analyses, would be helpful to visualize cellular trajectories objectively.

Response: We thank the reviewer for the intriguing question. As discussed in our response to the first reviewer, we have added analyses of somatic mutations and DNA copy number variations for the TNBC cell clusters.

We used Monovar [8] for detections of single nucleotide variation (SNV) from the SMART-seq reads of single cells. For the first TNBC study, the cells in C1 bear significantly more SNVs than the cells in C2. This is nicely consistent to our hypothesis that the cluster of C2 represents an early stage of tumor cells, whereas C1 represents a more malignant stage. By contrast, the cluster of C0, which mainly consists of the cells from micro-metastatic sites, exhibited slightly elevated rates of mutations, indicating that micro-metastasis takes places during early stage of tumor development. This is consistent to the previous notes [9, 10].

Finally, it should be noted that due to lack of non-cancerous cells from the same patient, these SNVs were identified by alignment of the RNA-seq reads with the human genome reference. They are not all somatic mutations. Inference of single-cell CNVs based on low-coverage RNA-seq data is quite challenging, and it requires non-cancerous cells as a baseline. We could not detect the CNVs of the cells in these three clusters. Nevertheless, the general trend of SNV numbers in the three clusters supports the proposed cellular patterns of tumor development revealed by SCASL based on the AS landscapes.

Numbers of SNVs in the single-cells inferred using MonoVar.

We have also performed trajectory analysis with two methods, Monocle and RNA velocity. The results of Monocle showed that the cells in C2 have the highest degree of differentiation, which is consistent to the result of CytoTRACE reported in the manuscript. However, the result of RNA velocity does not provide much insight into the trajectory of C2. As reported previously, RNA velocity is not applicable for such cases of cell clusters being discontinued from each other [18].

Trajectory analysis of the first TNBC study. (A) Pseudo-time analysis of the clustering results for the first TNBC dataset was performed using MONOCLE. T-tests were used to evaluate the statistical significance between different groups. *, <0.05; **, <0.01, ***, <0.001. **(B)** RNA velocity analysis of the cell clusters defined by SCASL.

7. For the second breast cancer dataset, Fig. 3B, why is there no UMAP from gene expression to make a comparison with AS, similar to Figs. 2A and B? It seems using only gene expression, in this case, would be enough to capture the C3 cluster. Similar to the above comment, how would the authors verify that the C3 cluster is the precancerous cells? The descriptive information from differential gene expression would not be enough. Would these precancerous have some additional information, e.g., the presence of mutations/copy number changes? More trajectory analysis results from different methods would be required. This would need supporting experimental validation to prove the link between the shift of splicing landscapes and the precancerous cells.

Response: As suggested by the reviewer, we have added the clustering results based on gene expression data (Fig. S5B, reproduced as follows). Indeed, the gene expression profile, in this case, can also classify the tumor and normal cells into clusters that are similar to the results of SCASL based on AS profiles. However, the expression-based clusters show much stronger batch effects than the AS-based results. In addition, in the AS-based tSNE plot, the C3 normal epithelial cell cluster is located in a position in-between the normal cell cluster C5 and other tumor cell clusters, which is highly indicative of its potential state as pre-cancer cells (Fig. 3A). By contrast, such insight is not readily available in the expression-based tSNE plot (Fig. S5B).

Clustering based on expression profile of TNBC-2. TSNE plot showing clustering of TNBC tumor cells based on expression profile, with labels indicating AS clusters from SCASL (left), cell types (middle), and patients (right).

As suggested by the reviewer, we have added analyses of somatic mutations and DNA copy number variations for the cell clusters defined by AS. As shown in Fig. S5E, the cells in C3 bear significantly more SNVs than the cells in C5. As expected, the tumor cells in C2 and C0 cells from the same patient bear more mutations than C3.

Next, we used inferCNV[19] to infer the DNA copy number changes for each single-cell, which are summarized as an cumulative CNV score. As shown in Fig. S5F, the cells in C3 bear intermediate levels of DNA copy number changes, which are higher than the normal cells in C5 but lower than the tumor cells in C2 and C0, a pattern highly consistent to that of the SNVs.

In summary, together with the signature genes reported in the previous version, the new observations of SNVs and CNVs further support that C3 indeed has gained tumorigenic features, not only in the gene expression profiles, but also in the landscapes of DNA mutations and copy number alterations.

Numbers of SNVs in the single-cells inferred using MonoVar. Copy number variation analysis of the single-cells was performed using inferCNV.

As suggested by the reviewer, we have also performed trajectory analysis with two methods, Monocle and RNA velocity. The results of Monocle showed that the cells in C3 and C5 are highly differentiated

(Fig. S5C), which is not surprising. After all, the cells in both clusters are normal epithelial cells. It is likely that even though the cells in C3 have gained tumorigenic features of gene expression, DNA mutation, and copy number variations, they are still at a non-malignant state. Again, the result of RNA velocity does not provide much insight into the trajectory of C5 and C3, which are discontinued from each other [18].

Trajectory analysis of TNBC-2. (A) Pseudo-time analysis of the AS profile clustering results in the TNBC-2 dataset for the same patient (PT89). T-tests were used to evaluate the statistical significance between different groups. *, <0.05; **, <0.01, ***, <0.001. **(B)** RNA velocity analysis of the cell clusters defined by SCASL.

8. There is no comparison between AS and gene expression for the datasets shown in Fig. 4a. Would the two cellular trajectories be reproducible using the gene expression? More objective visualization from multiple trajectory analyses would be required. Where were the arrows derived from in Fig. 4a? There is no description of what they mean in the legend.

Response: We have added the TSNE map based on gene expression data. As shown in Fig. S7A, the two major lineages of hepatocytes and cholangiocytes can be roughly classified based on gene expression data, which has been reported in the original study[20]. However, such a cluster distribution pattern does not reflect the time-dependent differentiation trajectory (Fig. S7B, C). The potential intermediate cell cluster between the lineages of hepatocytes and cholangiocytes is not apparent.

TSNE plot showing clustering of 447 embryonic liver cells by Seurat based on expression profiles. Cells are labeled according to the embryonic time (left), clusters defined by Seurat (middle), or clusters defined by SCASL based on AS (right).

As suggested by the reviewer, we are providing the results of Monocle and RNA velocity for cross-validations of the proposed differentiation lineages (Fig. S6A, C). The results of all three strategies, CytoTRACE, Monocle, and RNA velocity are highly consistent with each other.

(Left) Pseudo-time analysis of the AS clustering results was performed using MONOCLE. On the left are clusters of cells belonging to hepatoblasts and hepatocytes, and on the right are clusters of cells belonging to cholangiocytes. (Right) RNA velocity analysis of 447 embryonic liver cells clustered by SCASL based on AS. The arrows in RNA velocity indicate the direction and velocity of dynamic changes in gene expression.

We have added a new figure to show the embryonic days of the cells in the tSNE plot based on the AS profiles (Fig. S6B). The two arrows in Fig. 4A were simply put to reflect the time-dependent differentiation trajectories of the two lineages along the process of embryonic development.

9. There is a claim of reporting “a novel class of cells that potentially function in the crosstalk during embryonic liver development”. What it means by “crosstalk” in this context? Probably this needs to tone down since no experimental results support this. Would this C4 cluster be identified using gene expression?

Response: We agree that the claim of crosstalk sounds too strong. It is rather a speculation. The word “crosstalk” here refers to the fact that C4 has the AS and gene expression characteristics of both cholangiocytes and hepatocytes. In fact, transdifferentiation between cholangiocytes and hepatocytes in both directions has been well acknowledged [21-26]. Therefore, we suspect that this cluster represents a potentially intermediate stage between the two lineages. However, more direct evidence is needed to draw a firm conclusion. We have modified the text to tune down the claims here. We thank the reviewer for raising this comment.

As shown in Fig. S7A-C and discussed in our response above, the cell clusters defined by the gene expression profiles did not show clear trajectories or signs of potentially intermediate stages along or between the two lineages of hepatocytes and cholangiocytes.

10. It is not clear for the comparison of AS vs. expression UMAP in Fig. S7. How many AS and genes (variable features) and PCA dimensions were used? Were these UMAP generated in a fair comparison? For example, were they from the same genes? Also, it is difficult to visually inspect the comparison by looking at many labeled colors in UMAPs. This would need some other better plot types to summarize the comparison results. Would the Treg clusters (C3 and C4) be differentiated if used gene expression of the same genes as in AS?

Response: As mentioned in our previous response to comment #5, We have added the information of gene numbers used for the UMAP and correlation heatmaps into the Figure Notes and Methods section. In general, both the analyses based on AS and expression are transcriptome-wide and unbiased, and the lowly expressed genes were removed for both AS and expression analyses. Eventually, although the total number of genes used are slightly different in the AS and expression analyses, they are roughly the same group of genes with overlaps usually higher than 90%. Therefore, the comparisons between AS vs expression features are fair and reasonable.

It could be challenging to clearly mark the cell types in a dense UMAP plot. We have recolored the cell maps in Fig. S9 to better illustrate different cell clusters. In addition, we added a heatmap to show the overlaps between the two sets of cell clusters defined by AS and gene expression, respectively (Fig. S10).

Heatmap showing overlaps between the clusters defined based on AS or gene expression data of the T cells. The cell clusters defined by Seurat based on gene expression are arranged by columns, whereas the clusters defined by SCASL based on AS are arranged by rows. The color indicates the proportions of the overlapping cells in the expression clusters.

As shown in the following result of cell clustering based on the same group of genes used for AS clustering, C3 and C4 cells are not easily distinguishable. By contrast, these two cell clusters defined by SCASL are clearly separated on the AS map. Finally, it is worth noting that, as we are proposing, the AS heterogeneity could serve as the basis of a new scheme for definitions of cell subtypes, which show biological and physiological relevance. However, this does not diminish the insights generated by the expression-based cell clustering, which simply assess the cell heterogeneity from a different aspect.

UMAP visualizations showing the clustering results based on the expression profiles of genes present in the AS profile. The single-cells are labeled by the clustering results based on gene expression (left) or AS (right) profiles.

11. The manuscript describes a lot of descriptive information on T-cells analysis. This section is not concise, and some explanations are probably not informative and relevant to the manuscript, such as Fig. S10B.

Response: We agree that some descriptive information in the section about T cells is not directly relevant to the method and therefore distracting. As suggested by the reviewer, we modified this section to make it more concise and focused.

12. The concept that each single cell has just one dominant splicing form is not quite right. Recent studies using long-read sequencing revealed genes with multiple isoforms in a single cell (Volden et al., 2022; Tian et al., 2021; Mincarelli et al., 2023). The detection of isoforms per gene per single cell varies across genes and expression levels.

Response: We agree that the statement should be rephrased more carefully. The aforementioned studies [27-29] utilizing full-length sequencing highlighted the diversity of isoform subtypes within cell populations, without directly proving the absence of dominant isoform subtypes within individual cells. In fact, one example provided by [28] illustrates that most of the single-cells do express a dominant isoform (in Fig. S8C of [28]). However, we agree with the reviewer that detections of isoforms vary across genes with different expression levels. We have modified the text in the manuscript.

Isoforms of SRSF3 in CLL2, with UMAP visualization colored by two isoforms with differential expression across different clusters [28].

It is worth noting that the current second-generation sequencing technology does have limitations in extracting information of AS, especially for low-depth genes. However, studies have also demonstrated that errors stemming from the limitations of sequencing depth can be mitigated by implementing a filtering threshold for splice junction reads of greater than 10 [30], a criterion that aligns with our filtering threshold.

13. Is the AS probability matrix tolerant to the batch/donor effect? If not, how do the authors deal with this for the downstream analyses, e.g., clustering and differential splicing analysis?

Response: SCASL takes AS probabilities, instead of direct read counts, as input for clustering analysis. In theory, this should make SCASL less prone to batch effects in gene expression. However,

in cases where certain datasets possess notably robust donor effects, for example the second TNBC dataset we used, the clustering results may still be subject to batch-related influences (Fig. S5G). Nonetheless, accurately discerning whether this discrepancy arises from batch-related variations or genuine biological disparities proves challenging [31-33]. Consequently, for such datasets, as mentioned in the manuscript, we focused our downstream analyses on the clusters originating from the same patient.

Furthermore, if needed, users can take advantages of the popular batch removal techniques, including Combat [34], Scanorama [35], and others, to eliminate batch effects before running SCASL. However, as discussed rigorously in literature of the related field [31-33], it is worth noting that the process of batch removal can potentially lead to inadvertent elimination of genuine biological information. Therefore, we leave the choice of batch removal to the users of SCASL for their future studies.

14. In the discussion, it should be clearly stated that the SCASL cannot be used for the droplet-based single-cell data.

Response: We thank the reviewer for raising this comment. As discussed in our response to the first reviewer, we have tested the performance of SCASL on a dataset generated by 10x platform for dentate gyrus cells [7] (in mice at postnatal days 12, 16, 24, and 35). It is known that neuroblasts differentiate into granule cells. Clustering of neuroblasts and granule cells can be challenging, as both cell types undergo developmental transitions and share certain molecular markers. The cell clusters defined by SCASL based on 10x RNA-seq data indeed show patterns indicating potential differentiation transitions (Fig. S16A, B), which are supported by differentiation marker genes. By contrast, such transitions were not recapitulated by the cell clustered defined by their gene expression profiles, which only shows two major groups of neuroblasts and granule cells. It should be noted that the cell type annotations were defined based on gene expression profiles in the original study. Therefore, it is well expected to see the two types of cells clearly separated in Fig. S16C.

This analysis shows that despite the strong 3' bias, 10X RNA-seq data can also be used for cell clustering by SCASL. As discussed in the manuscript, the information of cell identities supplied by AS profiles are highly redundant, and therefore, fractions of the AS profiles within the 3' regions of mRNAs could be informative for definitions of cell clusters. Technically, the procedure of SCASL is exactly the same with either SMART-seq or 10X data. However, it should be noted that most of the 10X RNA-seq libraries were subjected to very low depth of sequencing. Therefore, splicing-based clustering with SCASL is not recommended for the droplet-based 3' end RNA-seq data with low sequencing depth.

Results of SCASL applied on 10X data of Dentate gyrus cells. (A) TSNE plot showing clustering of 2442 cells by SCASL based on the AS landscapes. The cells are color labeled by the clusters defined by SCASL (left) and cell types defined by marker genes (right). The principal component analysis (PCA) is performed with a dimensionality reduction number of 20. (B) The dot plot shows

the expression of marker and functional genes in poor differentiation (Neuroblast) and well differentiation (Granule cell). The dot sizes represent the proportions of cells with expression, and the color scale represents the mean expression level. (C) TSNE plot showing clustering of 2442 cells by Seurat based on the gene expression profiles with 3000 variable features. The cells are color labeled by the clusters defined by SCASL (left) and cell types defined by marker genes (right). The principal component analysis (PCA) is performed with a dimensionality reduction number of 20.

Minor points

1. Gene names in texts and figures must be in italics.

Response: We thanks the reviewer for the reminder. Gene names have been reformatted.

2. Table 1 shows only 5 datasets, not 6. A dataset from Ref. 93 seems to be missing from the table and main text.

Response: We have added the missing dataset into Table 1.

Table 1. Data resources of single-cell RNA-seq.

Cell source	Species	Method	Cell number	AS events	AS genes	Reference
TNBC	Homo sapiens	Smart-seq2	422	6111	2178	[36]
TNBC	Homo sapiens	Smart-seq2	443	3565	2248	[37]
Hepatoblast	Mus musculus	Smart-seq2	447	3204	1572	[20]
HCC Immune cells	Homo sapiens	Smart-seq2	2349	10257	3879	[38]
Brain	Mus musculus	Smart-seq2	1734	3058	1615	[39]
GBM Immune cells	Homo sapiens	Smart-seq2	1218	8235	2795	[40]
Brain	Mus musculus	10X	2442	267	170	[7]

3. TRDJ1 is missing in Fig. 5B.

Response: We are very sorry for this mistake. This issue has been fixed.

4. Why is the representation inconsistent across the manuscript, e.g., UMAP in Fig. 2 and TSNE in Figs. 3 and 4?

Response: We have replaced the UMAP plot in Fig. 2 with T-SNE plots. T-SNE is known for its capability of preserving local structures of the data [41]. However, for datasets with large sample

numbers, such as the T cell dataset and BEC dataset, we have opted to utilize UMAP, which has been shown to be more efficient in illustrating the global data structure for large-scale single-cell datasets [42, 43]. Shown below is a side-by-side comparison between T-SNE and UMAP plots with the same AS profiles of the T cells from HCC patients. We believe that the UMAP plot is a better illustration of the cell clusters defined by SCASL. Hence, we used T-SNE for relatively small datasets and UMAP for large datasets.

Visualizations of the T cells with T-SNE and UMAP. Cells in both figures are labeled according to the AS clusters.

5. Why two TSNE plots in S4 are different?

Response: We are sorry for not making this clear. Fig. S4A shows all the cells, including the epithelial cells, immune cells, and mesenchymal cells, while Fig. 3A and S4B were just for the epithelial cells (normal and TNBC tumor cells).

6. The correlation scale should be consistent across graphs. (e.g., Figs 2E, F and 3B, C, and others).

Response: First, there is an inherent difference in the distributions of the correlation coefficients obtained with the profiles of splicing and expression in different datasets. This is due to the different sample numbers, different feature numbers after filtering, and distinct distribution scales of the AS and expression profiles across datasets. Second, the correlation coefficients were meant to be compared only within the same similarity matrix. We do not compare different similarity matrices. Third, labeling the AS and expression similarities with different color tones makes the figures easier to read. It helps conveying the message that the AS and expression landscapes reflect different aspects of cell heterogeneity. Because of the reasons above, we prefer to use different color tones for the two types of similarity matrices to better illustrate the patterns of correlation coefficients between single cells within the same dataset.

7. No statistics on how many AS events were identified and how many genes they belonged to. It would be useful to see the distribution of detected AS event types (e.g., exon skipping, 5' and 3' alternative splicing, etc.). How do distinct AS event types contribute to the clustering analysis?

Response: As mentioned in our previous responses, we now have added the information of AS event numbers and gene numbers into the Table 1. The reviewer has raised a very intriguing question. The strategy we used for quantifications of AS events does not rely on prior annotations of exons. Its results provide a comprehensive survey of different splicing patterns, including the canonical AS events (such as exon skipping, intron retention, mutually exclusive exons) as well as other splicing errors due to rare mutations or aberrant spliceosome. Therefore, SCASL defines cell clusters based on a global analysis of the AS landscapes across hundreds of single-cells. It is not straight forward to quantify how much each type of AS contributes to the clustering analysis. Answering this question needs a new pipeline. We think that it is beyond the scope of the current study but will be worth further investigation in future research. We have added a brief discussion about this.

8. The figures showing differential variance results (e.g., Figs. 5C, 6A, B, D) lack statistical information (e.g., p-values).

Response: We thank the reviewer for the reminder. We have added the p-value information (Figs. 5C, 6A, B, D).

Code and software review

Response: We thank the reviewer for testing the algorithm and providing the detailed feedbacks. We have provided more documents, including the *requirements.txt* file to specify the software environment for SCASL and the *README.md* file for detailed instructions to the usage of SCASL. Alternatively, users have the option to execute the software in Google Colab, which should resolve potential issues related to software version conflicts. We have also provided a simple test code and demo data in Colab at

<https://colab.research.google.com/drive/1FckmDDmAZblChaSB9pg6jHQEHQCNPFUd?usp=sharing>

1. There is a strong lack of documentation for the tool on how to produce the input, run the tool, and control the parameters. For example, the main documentation states that the tool uses BAM files as the initial input, but from running and inspecting the code, it's clear that it also requires computed junction sites derived from BAM files. Also, no details are provided on how to derive these junctions, and the default example run is not working without additional edits to the config file.

Response: We are sorry for not provided more detailed instructions for the algorithm. We have now updated the documents on GitHub. More specifically, for the question above, the junction reads used for clustering are automatically extracted from the BAM file by SCASL, which generates a junction file in a sub-folder named "junction" located within the folder of the BAM file. Users also have the flexibility to skip the step above and generate junction files with alternative junction extraction software, such as STAR. Details are provided in the updated package.

2. The interpretation of the results also lacks documentation, e.g., the meaning of the plots, result file

formats and values, etc. In case a person is interested in using intermediate files, it will be hard to figure out what each intermediate output means.

Response: We thank the reviewer for the reminder. In the README.md file on GitHub, we have provided more instructions about interpretations of the result files, plots, and intermediate files.

3. In the file “config/srr_demo.yaml” , it should be “junction : data/junction” not “junction : data/junc” and “label_file : data/label.csv” should be “label_file: data/Demo_label.csv” .

Response: Fixed. We thank the reviewer for pointing them out.

4. The installation of the software has a problem. The reviewer successfully installed the SCASL pipeline only on the Linux system using the Conda environment. However, it never success on macOS. This is likely because the required LeafCutter software cannot be installed in R (tested on different versions of R) on macOS. See the error below. The reviewer did not try it on the Windows system. Since SCASL relies on the LeafCutter, this would also be a problem for the SCASL.

Response: We thank the reviewer for pointing out this issue. We are aware of the problem of installing LeafCutter in R on macOS, and therefore, we have provided an instruction to extract junction reads without installing LeafCutter. Please see the README.md file. In brief, SCASL only uses a function of LeafCutter to extract and count the junction reads. The source code for this function is provide in the leafcutter file (credit of LeafCutter acknowledged).

In addition, as mentioned in our previous responses, users also have the flexibility to skip the step above and generate junction files with alternative junction extraction software, such as STAR. Details are provided in the README.md file of the updated package. We have modified this part of the code on GitHub so that users can use junction files generated in various ways. We have included a trial run that takes junction files generated by STAR as input, in the provided Colab environment:

<https://colab.research.google.com/drive/1NGf7WKJEqTKhw1k84OebqMKRXEDbxMui?usp=sharing>

Reviewer #3 (Remarks to the Author):

I co-reviewed this manuscript with one of the reviewers who provided the listed reports as part of the Nature Communications initiative to facilitate training in peer review and appropriate recognition for co-reviewers

Reviewer #4 (Remarks to the Author):

I co-reviewed this manuscript with one of the reviewers who provided the listed reports as part of the Nature Communications initiative to facilitate training in peer review and appropriate recognition for co-reviewers

We thank the reviewers for their time and efforts and for their thoughtful comments.

Reviewer #5 (Remarks to the Author):

Xiang and colleagues presented a novel single-cell alternative splicing analysis method, that capable of conducting single-cell clustering based on alternative splicing and identifying previously undiscovered subgroups that are not detectable through single-cell gene expression clustering. Initially, the probabilities of the 5' and 3' ends were calculated as indicators of the degree of alternative splicing (AS). Subsequently, an iterative kNN (k-Nearest Neighbors) approach was employed to obtain AS probabilities for events with low sequencing depth and dropout events. These imputed AS probabilities were marked to reflect the influence of gene expression to a certain extent. Finally, a cluster analysis based on spectral clustering was conducted on the resulting matrix to delineate the alternative splicing landscape of single cells. Applying the SCASL approach to five datasets pertaining to disease and development, a series of novel subpopulations associated with alternative splicing were identified.

However, I have some concerns about certain aspects of the manuscript. In scRNA-seq clustering based on alternative splicing, three main challenges exist: low sequencing depth, dropout events, and interference from gene expression levels (with or without expression in the cell). While SCASL addresses the impact of sequencing depth and dropout events through imputation, it does not explicitly explain how to mitigate the influence of gene expression.

Main concerns:

1. The author employs the iterative kNN method to impute AS probabilities; however, there are concerns regarding the calculation indicator used for kNN. It appears that the Euclidean distance is calculated based on the probabilities of all AS events in cells, but it is crucial to consider whether the calculation indicator should be based on the probabilities of all AS events of the gene in the cell, rather than all genes.

Response: The reviewer has raised an intriguing question. We used all AS events for calculations of cell-cell distances because of the following reasons. 1) In general, the strategy of imputation assumes that a missing value of a particular cell can be inferred from the known values of other highly similar cells. Identifications of such cell groups should be based on global assessments of cell-cell similarities. In fact, it is a common practice to use the full transcriptome profiles for evaluations of cell-cell similarities, based on which missing values are inferred in parallel (referred to as multiple imputation). Such strategies have illustrated high accuracy and robustness for single-cell gene expression imputation, such as MAGIC [16] (Markov Affinity-based Graph Imputation of Cells), scImpute [17], SAVER [44] (Single-cell Analysis Via Expression Recovery), Drlmpute [45] (Dropout Regularization Imputation), etc. 2) Technically, the AS matrices are too sparse to make imputations based on the AS events of a single gene. For most of the genes, the low sequencing depth of single-cell RNA-seq simply does not generate enough junction reads covering multiple AS events of a gene. Therefore, imputation based on “local” AS similarities is not technically feasible and possible.

2. The clustering is directly performed based on the imputation probabilities and indicator vectors, which may not be optimal due to NA values in the AS probabilities matrix for genes not expressed in

the cell. To address this, the author should construct the real AS probabilities matrix using both NA values and non-NA values (AS probabilities). Additionally, clustering based on cell-gene NA values and non-NA values could resolve the heterogeneity present in cells, as demonstrated in the study "Embracing the dropouts in single-cell RNA-seq analysis, 2019". Therefore, the author needs to address three key problems: firstly, how to construct the real AS probabilities matrix; secondly, how to calculate the proximity distance between cells based on the AS probabilities matrix under such complex situations; and lastly, how to distinguish the influence of gene expression from alternative splicing during clustering based on the AS probabilities matrix in complex scenarios.

Response:

1) First, we think there is a misunderstanding. We did exactly what the reviewer has suggested. SCASL is based on the full AS matrices including both the original non-NA AS probabilities and the imputation values for the originally NA values. We did not only use the imputation probabilities. We are sorry for not making this clear. The text has been modified to prevent such misunderstanding. We believe that this has addressed the first key question raised by the reviewer.

2) For the second question, how to calculate the proximity distances, we applied spectral clustering for identifications of cells with high similarity. As a graph-based clustering algorithm [46], spectral clustering constructs a graph Laplacian matrix, performs eigenvalue decomposition to obtain a low-dimensional space, and then applies k-means clustering for the samples based on their proximity (squared Euclidean distances) in the low-dimensional space. It leverages the spectral properties of data to identify clusters, making it effective for datasets with complex structures or non-convex clusters. Spectral clustering has been shown to be efficient and robust on large-scale, polarized data like the AS matrices [47, 48].

3) For the third question, we totally agree with the viewpoint of the literature highlighted by the reviewer [49]. Dropouts present in single-cell data encompass crucial information, and thus, SCASL prepares a NA indicator matrix (akin to the notion of dropout binarization in "Embracing the dropouts in single-cell RNA-seq analysis"), which is then concatenated with the AS probability matrix, followed by spectral clustering of the cells.

The NA indicator (missingness matrix) indicates the number and positions of the missing values of splicing junction probabilities, which were shown as NA in the input data. Therefore, the missingness matrix (also known as the NA indicator matrix) indeed reflects the overall RNA abundance to some extent, albeit still being different from the gene expression profile.

As suggested by the reviewer, we tested the performance of SCASL without the information of missingness, by either completely removing the missingness matrix or shuffling the binary missingness values independently for each cell. As the reviewer has suspected, both operations resulted in significantly different results of cell clustering (Fig. S14, also reproduced as follows). For most of the datasets, removing or shuffling the missingness matrix would largely disrupt the cell clusters that we obtained before with SCASL. These new clusters fail to recapitulate important cellular heterogeneities between tumor vs normal cells, primary vs metastatic cells, and different cell development lineages. Therefore, the missingness matrix is indispensable for definitions of cell clusters by SCASL based on AS.

In addition, it is worth noting that the NA shuffling process above does not alter the number of the missing values in each cell. Therefore, the clustering results demonstrate that the positions of NAs,

rather than their numbers (i.e., overall sparseness), contributes to the clustering of cells.

On the other hand, the missingness matrix alone is certainly not informative enough to define the cell clusters. The percentages of missing AS probability values (values of NA) are comparable across the cell clusters defined by SCASL (Fig. S15, also reproduced as follows). Furthermore, upon shuffling of the AS probability matrix, SCASL also failed to recover the cell clusters with biological relevance (Fig. S14, reproduced as follows). Therefore, a critical and necessary feature of SCASL is to incorporate both the AS profile and the data missingness information for defining cell clusters. In other words, both the information of AS probabilities and data sparsity pattern contributes to the cell heterogeneity landscapes as portrayed by SCASL. The results above have been added into the revised manuscript.

Clustering with only AS information

Clustering results using only the AS information. For each of the 6 datasets, the figure in the top row shows the clustering results from the complete process of SCASL (AS and NA information), and the figures in the bottom row shows the results of using only the AS information.

Figures on the left show the clustering results using the complete process of SCASL (AS and NA information). Figures in the middle show the clusters defined by SCASL with the AS profile shuffled. Figures on the right show the clusters defined by SCASL with the missing data information shuffled.

The percentages of missing AS probability values (NA values) across the cell clusters defined by SCASL are shown in box plots.

3. Regarding batch effects, it would be valuable to understand how the author perceives their impact and how SCASL accounts for them during clustering based on alternative splicing.

Response: SCASL takes AS probabilities, instead of direct read counts, as input for clustering analysis. In theory, this should make SCASL less prone to batch effects in gene expression. However, in cases where certain datasets possess notably robust donor effects, for example the second TNBC dataset we used, the clustering results may still be subject to batch-related influences (Fig. S5G). Nonetheless, accurately discerning whether this discrepancy arises from batch-related variations or genuine biological disparities proves challenging [30-32]. Consequently, for such datasets, as mentioned in the manuscript, we focused our downstream analyses on the clusters originating from the same patient.

Furthermore, if needed, users can take advantages of the popular batch removal techniques, including Combat [33], Scanorama [34], and others, to eliminate batch effects before running SCASL. However, as discussed rigorously in literature of the related field [30-32], it is worth noting that the process of batch removal can potentially lead to inadvertent elimination of genuine biological information. Therefore, we leave the choice of batch removal to the users of SCASL for their future studies.

4. There are a few specific questions that require clarification. For instance, further explanation is needed on the calculation of the AS probabilities matrix and the construction of a $4N * M$ matrix for clustering. Additionally, the roles of upstream-related grouping, downstream-related grouping, and indicator values in the $4N * M$ matrix during clustering need elaboration.

Response: We thank the reviewer for pointing this out. This is indeed a misstatement. We have revised this paragraph to clarify the technical questions and provide a clearer description of the data-processing procedures (page 29).

5. Previous studies have demonstrated that the alternative splicing state of T cells has limited efficacy in distinguishing between CD8 and CD4 T cells. The question arises whether SCASL incorporates additional factors, such as gene expression, that contribute to their complete separation (Figure 5A and Figure S8B).

Response: We thank the reviewer for the insightful comment. As discussed in our response to comment #2, both the AS profile and the data sparsity pattern contributed to classifications of cell clusters by SCASL. In fact, based on literature and our analyses, the CD8 and CD4 T cells do have quite different patterns of AS events. Significant differences have been observed among specific subtypes of CD4+ and CD8+ T cells, including:

CD45 (PTPRC), a transmembrane protein tyrosine phosphatase, plays a crucial role in T cell receptor (TCR) signaling. Alternative splicing of CD45 generates distinct isoforms that exert unique effects on T cell activation and differentiation. CD4+ T cells predominantly express CD45RO. CD8+ peripheral T cells do not express CD45RO but have nine-fold higher expression of CD45RBC in comparison to CD4+ peripheral T cells [50].

CTLA-4 serves as a negative regulator of T cell activation and is important for immune tolerance.

Alternative splicing of CTLA-4 produces two major isoforms: a membrane-bound form (CTLA-4) and a soluble form (sCTLA-4). Studies have revealed that CD4⁺ T cells exhibit comparable expression levels of the two isoforms, CTLA-4 and sCTLA-4. In contrast, CD8⁺ cells display nearly 2.5 times higher expression of the full-length CTLA-4 compared to sCTLA-4[51].

Furthermore, we selected two clusters of tumor-infiltrating CD4⁺ and CD8⁺ T cells as examples. These two clusters are marked by a number of differential splicing events. Herein, we present the top ten differential splicing events observed in this analysis. Therefore, based on our analyses reported in the present study, we believe that the AS landscapes are indeed quite informative for separating the CD8 and CD4 T cells and further defining biologically relevant subtypes of T cells.

A list of differential splicing events between CD4 T cells (C3) and C8 T cells (C6). For each AS event, the dot size represents the proportion of cells in which the splicing probability was detectable from the RNA-seq reads, whereas the color scale represents the average AS probability in these cells. P-values are shown on the right side of each row.

References

1. Stekhoven, D.J. and P. Bühlmann, *MissForest—non-parametric missing value imputation for mixed-type data*. *Bioinformatics*, 2012. **28**(1): p. 112-118.
2. Templ, M., et al., *VIM: visualization and imputation of missing values*. 2021.
3. Troyanskaya, O., et al., *Missing value estimation methods for DNA microarrays*. *Bioinformatics*, 2001. **17**(6): p. 520-525.
4. Mertes, C., et al., *Detection of aberrant splicing events in RNA-seq data using FRASER*. *Nat Commun*, 2021. **12**(1): p. 529.
5. Wen, W.X., A.J. Mead, and S. Thongjuea, *MARVEL: an integrated alternative splicing analysis platform for single-cell RNA sequencing data*. *Nucleic Acids Research*, 2023. **51**(5): p. e29-e29.
6. Olivieri, J.E., R. Dehghannasiri, and J. Salzman, *The SpliZ generalizes ‘percent spliced in’ to reveal regulated splicing at single-cell resolution*. *Nature methods*, 2022. **19**(3): p. 307-310.

7. Hochgerner, H., et al., *Conserved properties of dentate gyrus neurogenesis across postnatal development revealed by single-cell RNA sequencing*. Nature neuroscience, 2018. **21**(2): p. 290-299.
8. Zafar, H., et al., *Monovar: single-nucleotide variant detection in single cells*. Nature methods, 2016. **13**(6): p. 505-507.
9. Vincent-Salomon, A., F.-C. Bidard, and J.-Y. Pierga, *Bone marrow micrometastasis in breast cancer: review of detection methods, prognostic impact and biological issues*. Journal of clinical pathology, 2008. **61**(5): p. 570-576.
10. MacDonald, I.C., A.C. Groom, and A.F. Chambers, *Cancer spread and micrometastasis development: quantitative approaches for in vivo models*. Bioessays, 2002. **24**(10): p. 885-893.
11. Huang, Y. and G. Sanguinetti, *BRIE: transcriptome-wide splicing quantification in single cells*. Genome biology, 2017. **18**: p. 1-11.
12. Song, Y., et al., *Single-cell alternative splicing analysis with expedition reveals splicing dynamics during neuron differentiation*. Molecular cell, 2017. **67**(1): p. 148-161. e5.
13. Shen, S., et al., *rMATS: robust and flexible detection of differential alternative splicing from replicate RNA-Seq data*. Proceedings of the National Academy of Sciences, 2014. **111**(51): p. E5593-E5601.
14. Najar, C.F.B.A., et al., *Identifying cell state-associated alternative splicing events and their coregulation*. Genome research, 2022. **32**(7): p. 1385-1397.
15. Li, Y.I., et al., *Annotation-free quantification of RNA splicing using LeafCutter*. Nature genetics, 2018. **50**(1): p. 151-158.
16. Van Dijk, D., et al., *Recovering gene interactions from single-cell data using data diffusion*. Cell, 2018. **174**(3): p. 716-729. e27.
17. Li, W.V. and J.J. Li, *An accurate and robust imputation method scImpute for single-cell RNA-seq data*. Nature communications, 2018. **9**(1): p. 997.
18. Gulati, G.S., et al., *Single-cell transcriptional diversity is a hallmark of developmental potential*. Science, 2020. **367**(6476): p. 405-411.
19. Tirosh, I., et al., *Dissecting the multicellular ecosystem of metastatic melanoma by single-cell RNA-seq*. Science, 2016. **352**(6282): p. 189-196.
20. Yang, L., et al., *A single - cell transcriptomic analysis reveals precise pathways and regulatory mechanisms underlying hepatoblast differentiation*. Hepatology, 2017. **66**(5): p. 1387-1401.
21. Russell, J.O., et al., *Hepatocyte - specific β - catenin deletion during severe liver injury provokes cholangiocytes to differentiate into hepatocytes*. Hepatology, 2019. **69**(2): p. 742-759.
22. Manco, R., et al., *Reactive cholangiocytes differentiate into proliferative hepatocytes with efficient DNA repair in mice with chronic liver injury*. Journal of hepatology, 2019. **70**(6): p. 1180-1191.
23. Okabe, H., et al., *Wnt signaling regulates hepatobiliary repair following cholestatic liver injury in mice*. Hepatology, 2016. **64**(5): p. 1652-1666.
24. Tarlow, B.D., et al., *Bipotent adult liver progenitors are derived from chronically injured mature hepatocytes*. Cell stem cell, 2014. **15**(5): p. 605-618.
25. Michalopoulos, G.K., L. Barua, and W.C. Bowen, *Transdifferentiation of rat hepatocytes into biliary cells after bile duct ligation and toxic biliary injury*. Hepatology, 2005. **41**(3): p. 535-544.

26. Pu, W., et al., *Bipotent transitional liver progenitor cells contribute to liver regeneration*. Nature Genetics, 2023. **55**(4): p. 651-664.
27. Volden, R. and C. Vollmers, *Single-cell isoform analysis in human immune cells*. Genome Biology, 2022. **23**(1): p. 1-21.
28. Tian, L., et al., *Comprehensive characterization of single-cell full-length isoforms in human and mouse with long-read sequencing*. Genome biology, 2021. **22**(1): p. 1-24.
29. Mincarelli, L., et al., *Single-cell gene and isoform expression analysis reveals signatures of ageing in haematopoietic stem and progenitor cells*. Communications Biology, 2023. **6**(1): p. 558.
30. Buen Abad Najar, C.F., N. Yosef, and L.F. Lareau, *Coverage-dependent bias creates the appearance of binary splicing in single cells*. Elife, 2020. **9**: p. e54603.
31. Nyamundanda, G., et al., *A novel statistical method to diagnose, quantify and correct batch effects in genomic studies*. Scientific reports, 2017. **7**(1): p. 10849.
32. Goh, W.W.B., W. Wang, and L. Wong, *Why batch effects matter in omics data, and how to avoid them*. Trends in biotechnology, 2017. **35**(6): p. 498-507.
33. Cai, H., et al., *Identifying differentially expressed genes from cross-site integrated data based on relative expression orderings*. International Journal of Biological Sciences, 2018. **14**(8): p. 892.
34. Johnson, W.E., C. Li, and A. Rabinovic, *Adjusting batch effects in microarray expression data using empirical Bayes methods*. Biostatistics, 2007. **8**(1): p. 118-127.
35. Hie, B., B. Bryson, and B. Berger, *Efficient integration of heterogeneous single-cell transcriptomes using Scanorama*. Nature biotechnology, 2019. **37**(6): p. 685-691.
36. Davis, R.T., et al., *Transcriptional diversity and bioenergetic shift in human breast cancer metastasis revealed by single-cell RNA sequencing*. Nature cell biology, 2020. **22**(3): p. 310-320.
37. Karaayvaz, M., et al., *Unravelling subclonal heterogeneity and aggressive disease states in TNBC through single-cell RNA-seq*. Nature communications, 2018. **9**(1): p. 1-10.
38. Zhang, Q., et al., *Landscape and dynamics of single immune cells in hepatocellular carcinoma*. Cell, 2019. **179**(4): p. 829-845. e20.
39. Tasic, B., et al., *Adult mouse cortical cell taxonomy revealed by single cell transcriptomics*. Nature neuroscience, 2016. **19**(2): p. 335-346.
40. Darmanis, S., et al., *Single-cell RNA-seq analysis of infiltrating neoplastic cells at the migrating front of human glioblastoma*. Cell reports, 2017. **21**(5): p. 1399-1410.
41. Van der Maaten, L. and G. Hinton, *Visualizing data using t-SNE*. Journal of machine learning research, 2008. **9**(11).
42. Becht, E., et al., *Dimensionality reduction for visualizing single-cell data using UMAP*. Nature biotechnology, 2019. **37**(1): p. 38-44.
43. Kobak, D. and P. Berens, *The art of using t-SNE for single-cell transcriptomics*. Nature communications, 2019. **10**(1): p. 5416.
44. Huang, M., et al., *SAVER: gene expression recovery for single-cell RNA sequencing*. Nature methods, 2018. **15**(7): p. 539-542.
45. Gong, W., et al., *Drlmpute: imputing dropout events in single cell RNA sequencing data*. BMC bioinformatics, 2018. **19**: p. 1-10.
46. Jordan, M.I. and Y. Weiss, *On spectral clustering: Analysis and an algorithm*. Advances in

- neural information processing systems, 2002. **14**: p. 849-856.
47. Von Luxburg, U., *A tutorial on spectral clustering*. Statistics and computing, 2007. **17**: p. 395-416.
 48. Jia, H., et al., *The latest research progress on spectral clustering*. Neural Computing and Applications, 2014. **24**: p. 1477-1486.
 49. Qiu, P., *Embracing the dropouts in single-cell RNA-seq analysis*. Nature communications, 2020. **11**(1): p. 1169.
 50. McNeill, L., et al., *CD45 isoforms in T cell signalling and development*. Immunology letters, 2004. **92**(1-2): p. 125-134.
 51. Pawlak, E., et al., *The soluble CTLA-4 receptor: a new marker in autoimmune diseases*. ARCHIVUM IMMUNOLOGIAE ET THERAPIAE EXPERIMENTALIS-ENGLISH EDITION-, 2005. **53**(4): p. 336.

Reviewer #1 (Remarks to the Author):

The authors have done a thorough job assessing my comments and concerns. I recommend acceptance.

Reviewer #2 (Remarks to the Author):

The manuscript has been improved. The authors addressed all of the major concerns. This reviewer has only minor points.

1. Gene names in texts and figures must be in italics. The author reformatted gene names in texts BUT NOT in figures (e.g., Figs 3, 5, 6, and in multiple supplemental figures).
2. As far as I know, BRIE is for "single-cell," not for "bulk" data.
3. Fig. 2D, "Repressed in C2 VS C1 & C3" should be "Repressed in C2 VS C1 & C0"
4. In the method section, it said, "Eight sets of single-cell RNA-seq data." Table 1 shows only seven sets of single-cell RNA-seq data.

Reviewer #2 (Remarks on code availability):

The authors fixed issues and addressed all of concerns about the software.

Reviewer #3 (Remarks to the Author):

I co-reviewed this manuscript with one of the reviewers who provided the listed reports as part of the Nature Communications initiative to facilitate training in peer review and appropriate recognition for co-reviewers.

Reviewer #4 (Remarks to the Author):

I co-reviewed this manuscript with one of the reviewers who provided the listed reports as part of the Nature Communications initiative to facilitate training in peer review and appropriate recognition for co-reviewers.

Reviewer #5 (Remarks to the Author):

The authors have addressed most of the issues. While some concerns still persist, mainly regarding the broad applicability and the validation of the method. This paper is ready for publication as a method.

Responses to the reviewer's comments

Reviewer #2 (Remarks to the Author):

The manuscript has been improved. The authors addressed all of the major concerns. This reviewer has only minor points.

1. Gene names in texts and figures must be in italics. The author reformatted gene names in texts BUT NOT in figures (e.g., Figs 3, 5, 6, and in multiple supplemental figures).
2. As far as I know, BRIE is for "single-cell," not for "bulk" data.
3. Fig. 2D, "Repressed in C2 VS C1 & C3" should be "Repressed in C2 VS C1 & C0"
4. In the method section, it said, "Eight sets of single-cell RNA-seq data." Table 1 shows only seven sets of single-cell RNA-seq data.

Response: We thank the reviewer for pointing out these issues. We have modified the figures and texts accordingly, to address all the 4 points above.